# LOCAL LOSS OPTIMIZATION IN THE INFINITE WIDTH: STABLE PARAMETERIZATION OF PREDICTIVE CODING NETWORKS AND TARGET PROPAGATION

**Satoki Ishikawa**[1,2]**, Rio Yokota**[1]**, Ryo Karakida**[2]
[1] Science Tokyo   [2] AIST
{ishikawa, rioyokota}@rio.scrc.iir.isct.ac.jp, karakida.ryo@aist.go.jp

## ABSTRACT

Local learning, which trains a network through layer-wise local targets and losses, has been studied as an alternative to backpropagation (BP) in neural computation. However, its algorithms often become more complex or require additional hyperparameters because of the locality, making it challenging to identify desirable settings in which the algorithm progresses in a stable manner. To provide theoretical and quantitative insights, we introduce the maximal update parameterization ($\mu$P) in the infinite-width limit for two representative designs of local targets: predictive coding (PC) and target propagation (TP). We verified that $\mu$P enables hyperparameter transfer across models of different widths. Furthermore, our analysis revealed unique and intriguing properties of $\mu$P that are not present in conventional BP. By analyzing deep linear networks, we found that PC's gradients interpolate between first-order and Gauss-Newton-like gradients, depending on the parameterization. We demonstrate that, in specific standard settings, PC in the infinite-width limit behaves more similarly to the first-order gradient. For TP, even with the standard scaling of the last layer, which differs from classical $\mu$P, its local loss optimization favors the feature learning regime over the kernel regime.

## 1 INTRODUCTION

Deep learning has achieved remarkable performance by building upon the backpropagation (BP) algorithm and developing architectures specialized for it (Rumelhart et al., 1986; LeCun et al., 1998; 2015). BP, however, is not always a suitable method for more general objectives, such as biologically plausible computation (Lillicrap et al., 2020; Bredenberg et al., 2024) or efficient distributed computation (Amid et al., 2022). A representative alternative is local loss optimization, a type of credit assignment problem, in which loss functions are defined layer-wise, and targets are set locally. The basic formulation involves performing regression on target signals at each layer to reduce the global error across the entire network: Predictive Coding networks, usually referred to as PC, generate their targets through the internal dynamics of inference (Whittington & Bogacz, 2017; Song et al., 2020; Salvatori et al., 2023), while Target Propagation (TP) generates them using feedback networks (Bengio, 2014; Lee et al., 2015; Ernoult et al., 2022).

In many cases, the use of local losses requires additional hyperparameters (HPs) and their careful tuning, making the algorithm configuration significantly more complicated compared to that of BP. For example, PC requires not only the usual HPs, such as learning rate and initialization of weight parameters but also those for the inference phase, such as the initialization of the state and the number of inference sequences. These HPs are primary considerations and have been reported as critical for ensuring stable training behavior (Pinchetti et al., 2024; Alonso et al., 2024; Rosenbaum, 2022). A few analyses have succeeded in providing theoretical intuition for such local learning algorithms by introducing specific conditions or additional corrections that bridge them to classical optimization formulations (Song et al., 2020; Alonso et al., 2022; Meulemans et al., 2020). However, such conditions are not always met in practice and may not be commonly shared across the entire family of methods. To develop local learning that is more easily manageable across a broader range of settings, it is promising to establish a theoretical foundation that enables the analysis of natural learning dynamics under fewer constraints.

For standard BP, deep learning theory offers insights into the universal properties of learning (Bahri et al., 2020; Bartlett et al., 2021). A key research focus in this area is understanding learning in the infinite-width limit, including studies on neural tangent kernel (NTK) and feature learning regimes (Jacot et al., 2018; Chizat et al., 2019; Mei et al., 2018; Bordelon & Pehlevan, 2022b). In particular, Yang & Hu (2021) provided a unified perspective on the parameterizations that realize these learning regimes and proposed maximal update parameterization ($\mu$P) as a unique scaling of HPs, such as random initialization and learning rates, that achieves feature learning in the infinite-width limit. Building on this developing theoretical foundation, we expect to gain universal insight into local learning, which has not yet been systematically analyzed.

In this work, we derive the $\mu$P for PC and TP and investigate hyperparameter transfer (the so-called $\mu$Transfer) across different widths. Although $\mu$P for SGD has been previously derived, the $\mu$P depends on the specific training algorithm, making it necessary to derive $\mu$P for each local learning algorithm. Our contributions are summarized as follows:

- While it is known that PC inference trivially reduces to gradient computation of BP under the fixed prediction assumption (FPA), a technical and heuristic condition, there is generally no guarantee that PC will reduce to BP, making it highly non-trivial to identify its $\mu$P. We first consider PC with a single sequential inference and reveal the $\mu$P even without FPA (Theorem 4.1). We also empirically verify the $\mu$Transfer of learning rates, showing that the optimal learning rate does not depend on the order of width.

- Second, for a more general context involving multiple inference sequences, we consider the convergence of the inference phase. We find that, for deep linear networks, we can explicitly obtain the local targets and losses at the fixed point of the inference, which depend on inference step sizes (Theorem 4.2). Interestingly, it takes a similar form to the Gauss-Newton (GN) gradient, but it can be reduced to the conventional first-order gradient descent (GD) depending on the parameterization and step sizes. We find that the eventual gradient is closer to GD for sufficiently wide neural networks under standard experimental settings with $\mu$P. We also confirm that a larger inference step size, identified through this analysis, enhances $\mu$Transfer of HPs.

- Finally, we derive $\mu$P for both TP and its variant difference target propagation (DTP) assuming linear feedback networks (Theorem 5.1). We reveal a distinct property that differs from BP and PC; the feedback network of (D)TP changes the preferable scale of the last layer compared to the usual $\mu$P and causes the absence of the kernel regime. In this sense, (D)TP favors feature learning more strongly than other learning methods.

Thus, this study provides a solid and qualitative foundation for the further development of local learning schemes in large-scale neural networks in the future.

## 2 RELATED WORK

**Local learning:** Most research on local learning stems from the exploration of biologically plausible learning (Lillicrap et al., 2020), with PC and TP following this line. As deep learning has evolved, local learning has also begun to focus on large-scale networks, and some models have achieved performances close to those trained with BP (Ernoult et al., 2022; Ren et al., 2023). Several algorithms are inherently structured to resemble the BP chain (Akrout et al., 2019) or to estimate first-order gradients (Scellier & Bengio, 2017). In contrast, PC relies on an inference phase, which essentially infers the appropriate activation values for hidden layers, and TP uses a feedback network, both of which are quite different from BP and seem to be fundamental designs for using local targets. However, their optimization properties are still not well understood. Alonso et al. (2022) proposed a modified PC as a proximal point algorithm (implicit SGD), though it requires additional corrections and adaptive rescaling (Alonso et al., 2024). Innocenti et al. (2023) proposed an inference phase computed by a GN method, but it requires a quadratic approximation of the local loss around a special initialization. As discussed in the next section, bridging to such classical optimization requires strong conditions that may deviate significantly from the original purpose and algorithm (Rosenbaum, 2022; Meulemans et al., 2020).

**Infinite width and $\mu$P:** While the NTK regime guarantees the existence of learning dynamics in the infinite-width limit and its global convergence, it reduces to just a kernel method (Jacot et al.,

2018; Lee et al., 2019). To realize feature learning in the infinite-width limit, Yang & Hu (2021) proposed $\mu$P, which is a non-trivial scaling of HPs with respect to the width. From a theoretical perspective, this serves as a parameterization that enables the dynamics of feature learning, such as those described by the mean-field regime (Mei et al., 2018) or the dynamical mean-field theory (Bordelon & Pehlevan, 2022b). For more applications, $\mu$P or its extension has been validated across various architectures (Yang et al., 2021; Vyas et al., 2023; Everett et al., 2024). It covers not only the naive first-order gradient but also entry-wise adaptive optimizers such as Adam (Yang & Littwin, 2023) and second-order optimization methods like K-FAC (Ishikawa & Karakida, 2024). There has been little previous work on the infinite-width analysis of local learning. Bordelon & Pehlevan (2022a) formulated (direct) feedback alignment and (supervised) Hebbian learning using dynamical mean-field theory, which are rather close to BP.

## 3 PRELIMINARIES

In this section, we summarize local learning and $\mu$P in an $L$-layer fully connected neural network $f$:

$$h_l = \phi(u_l), \quad u_l = W_l h_{l-1} \quad (l = 1, \ldots, L), \tag{1}$$

where $W_l \in \mathbb{R}^{M_l \times M_{l-1}}$ are weight matrices, $h_l, u_l \in \mathbb{R}^{M_l \times N}$ are activations and $N$ is the number of data samples, independent of the order of width $M_l$. We set the width of the hidden layers to $M_l = M$ for $(l = 1, \ldots, L - 1)$ for simplicity. To keep the notation concise, for non-linear networks, we set $M_L = 1$; however, we can easily generalize to $M_L = \Theta(1)$. The activation function $\phi(\cdot)$ is usually assumed to be differentiable and polynomially bounded for some theoretical reasons within the $\mu$P framework (Yang & Hu, 2021).

### 3.1 OVERVIEW OF LOCAL LEARNING

#### 3.1.1 PREDICTIVE CODING

Predictive Coding (PC) updates both the states and weights to minimize the following free-energy function (Whittington & Bogacz, 2017; Song et al., 2020; Salvatori et al., 2023):

$$\mathcal{F}(v, W) = \gamma_L \mathcal{L}(y, W_L \phi(v_{L-1})) + \sum_{l=1}^{L-1} \gamma_l \frac{1}{2} \|v_l - W_l \phi(v_{l-1})\|^2. \tag{2}$$

$\mathcal{L}$ denotes a loss function, and both mean squared error loss and cross-entropy loss are allowed in theory and experiments unless an explicit assumption is stated. To distinguish the internal state from the forward signal propagation $u_l$, we denote this state as $v_l$. Although this algorithm was originally derived from the variational Bayes formulation, it has been extended beyond the scope of the original framework, aiming instead to develop inference computations that work more effectively in practice. PC is composed of two phases: an inference phase, in which the per-layer states $v_l$ are updated and a learning phase, in which weights $W_l$ are updated. Its update rule for the inference phase is given by

$$v_{l,s+1} = v_{l,s} - \frac{\partial \mathcal{F}}{\partial v_l} = v_{l,s} - \gamma_l e_{l,s} + \gamma_{l+1} \phi'(v_{l,s}) \circ W_{l+1}^\top e_{l+1,s} \quad (l < L), \tag{3}$$

where we define $e_{l,s} := v_{l,s} - W_l \phi(v_{l-1,s})$ and $\circ$ is the Hadamard product. From eq. (3), $\gamma_l$ can be regarded as a step size for the inference phase. The update rule for the learning phase is given by

$$W_{l,t+1} = W_{l,t} - \eta_l \frac{\partial \mathcal{F}}{\partial W_l} = W_{l,t} + \eta_l \gamma_l e_{l,s} \phi(v_{l-1,s})^\top. \tag{4}$$

Note that the inference time index $s$ and the parameter update index $t$ are distinct with $s$ resetting to 0 at each $t$. We usually omit the step size $\gamma_l$ in Eq. (S.6) in implementation. Generally, weights are updated after multiple inference steps, while the incremental version of PC (iPC), which updates the weights after just a single inference, has also been proposed (Salvatori et al., 2024b). The internal state can be updated simultaneously across all layers or computed sequentially in a specified order. In the first part of the next section, we focus on the Sequential Inference (SI) method, where $e_{l,s}$ is computed sequentially by propagating from the output layer to the input layer. For more details on this difference, see Algorithm 1 in the Appendix.

Empirically, to improve trainability, PC often relies on assumptions that are either rational or, at times, unrealistic. One such reasonable assumption is the initialization method for $v_{l,0}$, which is used to improve the convergence (Song et al., 2020; Alonso et al., 2022; Rosenbaum, 2022):

**Technique (i): Forward initialization (F-ini).** At each training step $t$, $v_{l,0}$ is initialized such that $v_{l,0} = u_{l,t}$ , which ensures $e_{l,0} = 0$.

Generally, the gradient computation of PC does not match that of BP. However, under F-ini and SI, it reduces to BP by adopting the following rather technical assumption (Millidge et al., 2022b; Rosenbaum, 2022):

**Technique (ii): Fixed prediction assumption (FPA).** Replace $\phi(v_{l-1,s})$ with $\phi(v_{l-1,0})$ during the inference phase.

Under FPA, the inference is given by $e_{l,s+1} = (1 - \gamma_l)e_{l,s} + \gamma_{l+1}\phi'(v_{l,0}) \circ W_{l+1}^\top e_{l+1,s}$. By substituting F-ini, one can easily verify that this sequential inference computes $\nabla_{u_l}\mathcal{L}$. In Section 4, we reveal that the following scaling of $\gamma_L$ with respect to the width $M$ plays a fundamental role in characterizing the feature learning of PC and a parameterization that enables stable learning even without such heuristic techniques:

$$\gamma_L = \gamma'/M^{\bar{\gamma}_L} \tag{5}$$

with an exponent $\bar{\gamma}_L$ and an uninteresting constant $\gamma' > 0$. A more detailed overview of PC is provided in the extended related work (Appendix.A.1.2).

### 3.1.2 Target Propagation

In target propagation (TP), $\hat{h}_L = h_L - \hat{\eta}\nabla_{h_L}\mathcal{L}$ is propagated through the feedback network, which generates local targets $\hat{h}_l$ as follows:

$$\hat{h}_l = g_l(\hat{h}_{l+1}), \quad g_l(x) = \psi(Q_l x) \quad (l = 1, ..., L-1), \tag{6}$$

where $Q_l \in \mathbb{R}^{M_{l-1} \times M_l}$ are weight matrices, $\psi(\cdot)$ is an activation function of the feedback network. We also analyze the Difference Target Propagation (DTP), a variant of TP, whose definition is provided in the appendix. The feedback network is trained to minimize the following reconstruction loss:

$$\mathcal{L}_{\textbf{rec}}(Q_l) = \|g_l(f_l(h_{l-1})) - h_{l-1}\|^2, \tag{7}$$

where $f_l(x) = \phi(W_l x)$. TP updates the weights $W_l$ to minimize the following local loss $\|e_l\|^2 := \|\hat{h}_l - h_l\|^2$. The gradient of this local loss provides the update rule for the learning phase as $W_{l,t+1} = W_{l,t} - \eta_l \phi'(W_{l,t}h_{l-1}) \circ e_l h_{l-1}^\top$. For a so-called invertible network, TP computes the Gauss-Newton Target (GNT), i.e., $e_l^{\text{GNT}} = (\delta_l \delta_l^\top + \rho I)^{-1}\delta_l e_L$ where $\delta_l = \nabla_{u_l} u_L$ is the BP signal (Meulemans et al., 2020) [1] and $e_L = y - h_L$ is the error vector. Note that the assumption of the invertible network is restrictive because the invertible network requires invertible activation functions, regular weight matrices, and the training tp converge to the solution of $g_l(\hat{h}_{l+1}) = f_{l+1}^{-1}(\hat{h}_{l+1}) = W_{l+1}^{-1}\phi^{-1}(\hat{h}_{l+1})$. For general networks, (D)TP does not necessarily lead to the GNT.

**Remark on a connection between PC and TP.** Some previous studies have argued that PC yields GNT-like solutions, and thus can be connected to TP (Alonso et al., 2022; Millidge et al., 2022a). These works attempt to gain an intuitive insight from the fixed point equation for each layer:

$$h_l^* = (W_{l+1}^\top W_{l+1} + \gamma_{l+1}/\gamma_l I)^{-1}(W_{l+1}^\top h_{l+1}^* + \gamma_{l+1}/\gamma_l W_l h_{l-1}^*), \tag{8}$$

where $h_l^*$ means $\phi(v_l^*)$. For $\gamma_{l+1}/\gamma_l \ll 1$, we approximate $h_l^* \approx W_{l+1}^\dagger h_{l+1}^*$. If we multiply this approximation across layers, the naive expectation is that $h_l^* \approx \prod_{i=l+1}^L W_i^\dagger e_L^*$, which corresponds to the GNT for linear networks. Thus, we can intuitively see that the PC may be linked to the GNT, although its exact connection requires careful limit operations across layers. Additionally, taking the limits $\gamma_{l+1}/\gamma_l \ll 1$ for all layers means the exponential decay of $\gamma_l$ with depth, raising concerns regarding its practical relevance. For $\gamma_l = 1$, Innocenti et al. (2024) has recently derived an explicit formulation of the free energy at the fixed point using an unfolding calculation of a hierarchical Gaussian model. This formulation shows that the obtained gradient differs from that of the exact GNT, supporting the idea that the connection to GNT would be weak.

---

[1] The final gradient $d\mathcal{F}/dW_l$ is equivalent to the special case of K-FAC (Martens & Grosse, 2015) where the preconditioners are applied only to the backward signals.

**Table 1:** Parameterization for weight initialization scale: $b_l$ and learning rate scale: $c_l$. Predictive Coding (PC) with $\bar{\gamma}_L = 0$ reduces to SGD's $\mu$P, while one with $\bar{\gamma}_L = -1$ reduces to $\mu$P for Gauss Newton Target (GNT). TP (Target Propagation) has the distinctive property of $b_L = 1/2$.

| Layer | SP (Default) | SGD (2021) | GNT (2024) | **PC (New)** | **TP (New)** |
|---|---|---|---|---|---|
| Input | $(0, 0)$ | $(0, -1)$ | $(0, 0)$ | $(0, -\bar{\gamma}_L - 1)$ | $(0, 0)$ |
| Hidden | $(1/2, 0)$ | $(1/2, 0)$ | $(1/2, 1)$ | $(1/2, -\bar{\gamma}_L)$ | $(1/2, 1)$ |
| Output | $(1/2, 0)$ | $(1, 1)$ | $(1, 1)$ | $(1, 1)$ | $(1/2, 1)$ |

## 3.2 $\mu$P AND LEARNING REGIMES

The abc-parameterization $\{a_l, b_l, c_l\}_{1 \le l \le L}$ determines the scaling of weights and learning rates at initialization. It scales the parameters by width as follows (Yang & Hu, 2021):

$$W_l = w_l/M^{a_l}, \quad w_l \sim \mathcal{N}(0, \sigma'^2/M^{2b_l}), \quad \eta_l = \eta_l'/M^{c_l}. \tag{9}$$

$\mu$**P and its conditions**: Consider the temporal change of $u_l$ by the parameter update:

$$\Delta u_{l,t} := u_{l,t} - u_{l,0} = \Theta\left(1/M^{r_l}\right), \tag{10}$$

where $\Theta(\cdot)$ denotes the order with respect to the width and $x = \Theta(M^a)$ means $\sqrt{\|x\|^2/M} = \Theta(M^a)$ for $x \in \mathbb{R}^M$. The training dynamics and parameterization are referred to as *stable* when $u_{l,0}$ neither vanish nor explode as the network width increases and $\Delta h_{l,t}$ do not explode as the network width increases (Definition A.2). Yang & Hu (2021) introduced the following conditions and characterized $\mu$P as a unique stable abc-parameterization under them:

**Condition 3.1** ($W_l$ updated maximally). $\Delta W_{l,t} h_{l-1,t} = \Theta(1)$ *where* $\Delta W_{l,t} := W_{l,t} - W_{l,0}$.

**Condition 3.2** ($W_L$ initialized maximally). $W_{L,0} \Delta u_{L-1,t} = \Theta(1)$.

These conditions imply $r_l = 0$ for all layers and feature learning. In contrast to this feature learning regime, the previous work refers $r_{l<L} > 0$ and $r_L = 0$ as the kernel regime. The NTK parameterization corresponds to the kernel regime with $r_{l<L} = 1/2$ Note that the original derivation of the parameterization that satisfies the above conditions is based on the first (infinitesimal) one-step update of the parameters (Yang & Hu, 2021; Ishikawa & Karakida, 2024) (see Section A.2). Our work also follows the same approach.

$\mu$**P for Gauss-Newton Target:** The following work has recently derived the $\mu$P scaling, including both first-order and second-order optimizations.

**Proposition 3.3** (Ishikawa & Karakida (2024)). *Consider the first one-step update by the GNT:* $W_{l,1} = W_{l,0} - \eta_l \phi'(W_{l,t} h_{l-1}) \circ (\delta_l \delta_l^\top + \rho I)^{-e_B} \delta_l diag(e_L) h_{l-1}^\top$ *where* $\delta_l = \nabla_{u_l} u_L$ *and* $e_L = y - h_L$. *In the infinite-width limit, this update admits the $\mu$P for feature learning at*

$$\begin{cases} \theta_1 = e_B - 1, & \theta_{1<l<L} = e_B, & \theta_L = 1 \\ b_1 = 0, & b_{1<l<L} = 1/2, & b_L = 1, \end{cases} \tag{11}$$

*where* $\theta_l := 2a_l + c_l$. *We obtain $\mu$P of SGD for $e_B = 0$, and that of GNT for $e_B = 1$.*

More precisely, we can also allow $b_L \ge 1$ for the feature learning regime. However such initialization reduces to the case of $b_L = 1$ in the next parameter update. Thus, we can summarize it as $b_L = 1$. The scaling of $b_L = 1$ implies that a smaller initialization is required compared to the standard parameterization (SP), which is PyTorch's default, for sufficiently wide neural networks. It can also be immediately verified that we can set $a_l = 0$ due to shift invariance without loss of generality. In Table 1, we summarize the $\mu$P from previous work and our results obtained in the following sections.

# 4 FEATURE LEARNING OF PREDICTIVE CODING

## 4.1 $\mu$P OF PC WITH SINGLE-SHOT SEQUENTIAL INFERENCE

As noted in section 3.1.1, PC involves such techniques as F-ini, FPA and SI, which must be clearly distinguished when deriving the $\mu$P. It is well-established that when F-ini, SI, and FPA are all assumed,

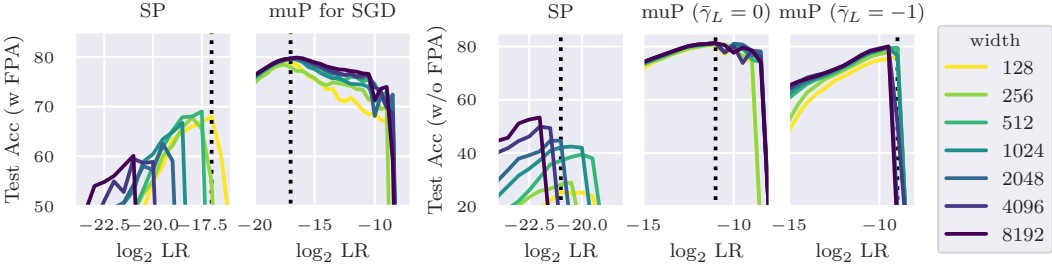

**Figure 1: $\mu$P enables the transfer of learning rates across widths.** (Left) PC reduces to SGD when F-ini, FPA, and SI are applied. In fact, using the $\mu$P of SGD, learning rates are successfully transferred across different widths. (Right) Even without FPA, our $\mu$P of PC also allows $\mu$Transfer across widths. In this case, inference is performed only once, and the difference in test accuracy between $\bar{\gamma}_L = 0$ and $\bar{\gamma}_L = 1$ is small. Both figures show results with a 3-layer MLP on FashionMNIST.

PC reduces to the gradient computation of BP, and the $\mu$P matches that of standard BP. Figure 1 shows that when F-ini, FPA, and SI are applied, the $\mu$P of BP can be directly transferred to PC and leads to learning rate transfer across width. However, this may not hold for general PC and BP as there is no guarantee of their equivalence. To explore this, we first remove FPA. Although initialization (F-ini) and sampling (SI) are inherently arbitrary, the justification for FPA is unclear from both machine learning and biological perspectives. In PC without FPA, we find the $\mu$P as follows:

**Theorem 4.1** ($\mu$P for PC (informal)). *Let the inference step sizes be $\gamma_{l<L} = \Theta(1)$ and $\gamma_L = \gamma'/M^{\bar{\gamma}_L}$ with a positive constant $\gamma'$. Consider the first one-step update of the learning parameters after a first single-shot SI with F-ini. Then, PC admits the $\mu$P for feature learning at*

$$
\begin{cases}
\theta_1 = -\bar{\gamma}_L - 1, \quad \theta_{1<l<L} = -\bar{\gamma}_L \geq 0, \quad \theta_{l=L} = 1, \quad (\theta_l = 2a_l + c_l) \\
b_1 = 0, \quad b_{l<L} = 1/2, \quad b_L = 1.
\end{cases}
\tag{12}
$$

*Rough sketch of the derivation.* Section B.1 of the Appendix presents a detailed and comprehensive derivation. It is based on the perturbation approach, which applies to general networks with nonlinear activation functions. This method is inspired by the previous work that derived the $\mu$P by evaluating Conditions 3.1 and 3.2 using the perturbations, such as $\partial_{\eta'}(\Delta W_{l,1} h_{l-1,1})\big|_{\eta'=0} = \Theta(1)$. This allows for a systematic and transparent derivation. In PC, we extend the perturbation argument to the inference step size and require

$$
\partial_{\gamma'}\partial_{\eta'}(\Delta W_{l,1} h_{l-1,1})\big|_{\eta'=\gamma'=0} = \Theta(1),
\tag{13}
$$

which is an example of Condition 3.1 for the hidden layer. Under the assumption of F-ini ($e_{l,0} = 0$), by putting $\delta_l = \nabla_{u_l} u_L$ ($l < L$) and $\delta_L = y - W_L v_{L-1,0}$, we obtain $u_{l,1} - u_{l,0} = -\prod_{i=l+1}^{L} \gamma_i \delta_l$, and

$$
e_{l,1} = \left(u_{l,0} - \prod_{i=l+1}^{L} \gamma_i \delta_l\right) - \phi\left(W_{l,0}\left(u_{l-1,0} - \prod_{i=l}^{L} \gamma_i \delta_{l-1}\right)\right).
\tag{14}
$$

For the hidden layers, the perturbation term (13) becomes $M^{-(\theta_l + \bar{\gamma}_L)}(-\delta_l + \phi'(W_l u_{l-1,0}) \circ W_l \delta_{l-1}) h_{l-1,0}^\top h_{l-1,0}$ and we obtain $\theta_l + \bar{\gamma}_L - 1 + (a_L + b_L) = 0$. We can similarly evaluate the other layers. The last condition, $b_L = 1$, comes from Condition 3.2. We can derive the NTK parameterization of PC in the same way.

As Figure 1 demonstrates, the obtained $\mu$P supports $\mu$Transfer in PC without FPA. Note that $\mu$Transfer is defined as satisfying both conditions: the optimal learning rate can be set independently of the order of width, and the empirical rule that 'wider is better' holds (Yang et al., 2021). This means that the optimal hyperparameters tuned for smaller-width models can be effectively re-used in larger-width models. Additionally, consistent with previous work, we observed the empirical rule of "wider is better" in $\mu$P (Yang et al., 2021), where test accuracy improves as the network width increases. The derivation of $\mu$P through a one-step update can be immediately generalized to cross-entropy loss in the same way as for $\mu$P of naive gradient descent. Thus, $\mu$Transfer can similarly be observed for cross-entropy loss, as shown in Appendix (Figure S.5).

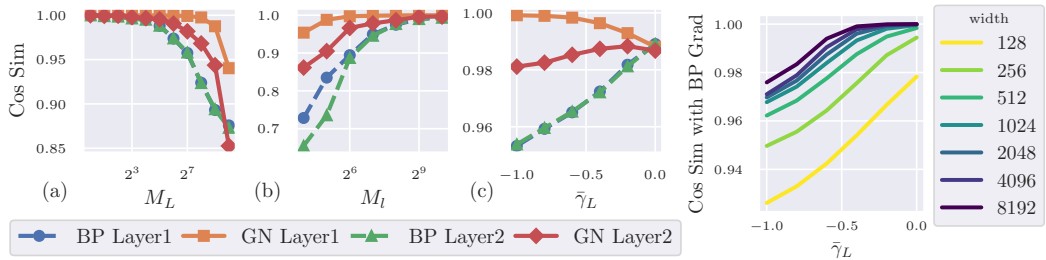

**Figure 2: (Left) Comparison of gradients with the analytical solution of a linear network.** We measured the cosine similarity between the gradients analytically derived in Theorem 4.2 and the BP gradients or GN gradients for each layer. (a) As $M_L$ approaches 1, PC's gradient converges to BP's. (b) As $M_l$ increases, the PC gradient approaches BP's. (c) $\bar{\gamma}_L = 0$ yields gradients closer to BP gradient (which means SGD in this experiment) compared to $\bar{\gamma}_L = -1$. **(Right) In a nonlinear MLP, PC's gradient also approaches BP's when $\bar{\gamma}_L = 0$.**

Thus far, when considering the parameter gradients, it appears that $\bar{\gamma}_L$ as a free parameter can be absorbed into the learning rate, allowing the feature learning dynamics to remain stable. However, as the following analysis shows, $\gamma_L$ modifies the preconditioning of the computed gradients, which may influence $\mu$Transfer of both $\gamma_L$ itself and the learning rate. The analysis also demonstrates the validity of setting $\gamma_{l<L} = \Theta(1)$.

## 4.2 ANALYSIS WITH LINEAR NETWORK

In the previous section, we derived the $\mu$P under the assumption that inference is performed only once using F-ini and SI. However, in practice, the inference phase typically involves multiple update steps. To address this, we found that it is possible to explicitly derive the following general solutions (fixed points) of the inference phase for linear networks. See Section B.2 for the derivation.

**Theorem 4.2.** *Suppose an $L$-layered linear network and a mean squared error loss $\mathcal{L}(y, W_L v_{L-1})$, and put $e_l^* = v_l^* - W_l v_{l-1}^*$, with $*$ denoting the fixed point of the inference process (3). The following holds:*

$$e_l^* = \frac{\gamma_L}{\gamma_l} W_{L:l+1}^\top (I + C_\gamma(W))^{-1}(W_{1:1}x - y), \quad C_\gamma(W) := \sum_{i=2}^{L} \frac{\gamma_L}{\gamma_{i-1}} W_{L:i} W_{L:i}^\top \quad (15)$$

$$v_l^* = W_{l:1}x + \left(\frac{\gamma_L}{\gamma_l} W_{L:l+1}^\top + \sum_{i=2}^{l-1} \frac{\gamma_L}{\gamma_{i-1}} W_{l:i} W_{L:i}^\top\right)(I + C_\gamma(W))^{-1}(y - f). \quad (16)$$

*and $e_L^* = y - W_L v_{L-1}^* = (I + C_\gamma(W))^{-1}(W_{1:1}x - y)$ where $W_{L:i} = W_L W_{L-1}...W_i$.*

From this general solution, we can also confirm the following property of the infinite width.

**Corollary 4.3.** *Suppose the setting of Theorem 4.2, $\gamma_{l<L} = \Theta(1)$ and the random weights given by $\mu$P i.e., $a_L + b_L = 1$. In the infinite-width limit, the PC's gradient reduces to the first-order GD for $\gamma_L = \Theta(1)$. For $\gamma_L = \Theta(M)$, the preconditioner part $C_\gamma$ remains of order 1.*

The exact solutions $e_l^*$ provide much clearer insight into the gradient computation compared to Eq. (8), which was previously argued but not explicitly solved. First, it becomes evident that PC does not generally coincide with GNT. Consequently, PC is also generally different from TP. In fact, PC coincides with GNT only for the input layer in a shallow network (i.e., $L = 2$), where the update vector for PC corresponds to a GNT update with a damping term. Although PC does not entirely coincide with GNT, it is noteworthy that the scaling of $c_l$ in $\mu$P for $\bar{\gamma}_L = -1$ matches that of GNT. In contrast, for $\bar{\gamma}_L = 0$, the PC's gradient aligns with the first-order GD. Because the preconditioner part scales as $C_\gamma = O(1/M)$ in the infinite-width limit, we observe that $e_l^* = \delta_l$, which reduces to the first-order GD. Naturally, $\mu$P matches that of first-order GD in this case. Intuitively, $\gamma_L$ reflects how effectively the last layer's error propagates downward. Like linear regression, the last layer's inference solution inherently involves an inverse matrix. Thus, when $\gamma_L$ is larger than other $\gamma_l$ values, the last layer's representation is computed first and propagated downward, making the solution resemble GNT.

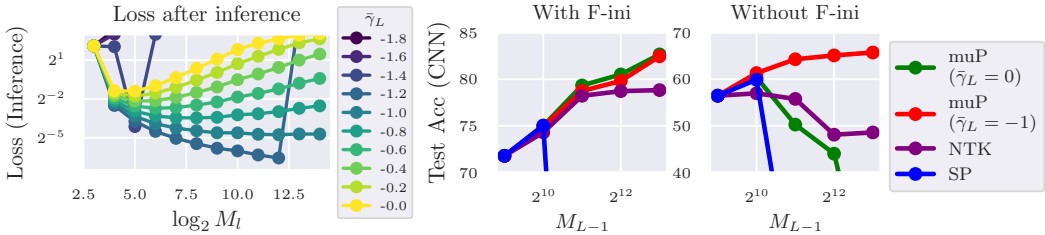

**Figure 3: (Left)** $\bar{\gamma}_L = -1$ **steadily reduces the local loss as width increases.** We observed the inference loss in a randomly initialized linear network for various $\bar{\gamma}_L$. For $\bar{\gamma}_L = -1$, the inference loss consistently decreases with increasing width. **(Right) The "wider is better" trend holds for $\mu$P with $\bar{\gamma}_L = -1$.** With F-ini, this trend holds for $\mu$P regardless of the $\bar{\gamma}_L$ value. However, without F-ini, the benefits of $\bar{\gamma}_L = -1$ become particularly prominent.

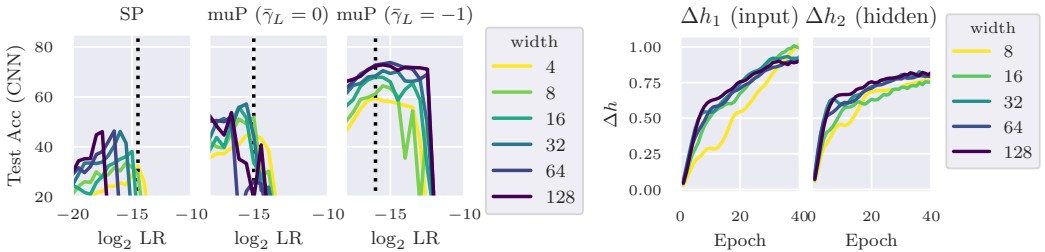

**Figure 4: (Left)** $\mu$P **can transfer the learning rate across widths (without F-ini).** We trained a 3-layer CNN on FashionMNIST with 100 inference iterations. Without F-ini, the stability of the inference becomes more crucial. As a result, unlike the single-shot SI with F-ini shown in Figure 1, the stability provided by $\bar{\gamma}_L = -1$ becomes critical. Note that additional experiments under different settings, including those with VGG5 (Figure S.3 ) and cross-entropy loss (Figure S.5 ), are presented in Section D.1.2 of the Appendix. **(Right)** $\Delta h$ **remains consistent across widths during training.** We confirm that the condition $\Delta h = \Theta(1)$ required by $\mu$P holds throughout the training.

Second, the order of $e_l^*$ in the analytical solution for the linear network matches the order of $e_{l,1}$ as derived in Theorem 4.1. Therefore, this theorem implies that in linear networks, the $\mu$P of PC would remain unchanged regardless of the presence of F-ini or the number of inference iterations. Moreover, as proved in Section B.2.3, the orders of $e_l^*$ and $e_{l,1}$ align only when $\gamma_{l<L} = \Theta(1)$. In practical settings with multiple inferences, it is desirable for the $\mu$P to be consistent both after a single inference and after the inference has fully converged. Therefore, setting $\gamma_{l<L} = \Theta(1)$ is reasonable.

Additionally, we found that the dimension of the last layer plays a key role in determining the similarity between PC and BP. According to the solution for linear networks, when $M_L = 1$, the PC's gradient aligns with GD. Figure 2 shows numerical results confirming that for $M_L = 1$, the gradient direction always corresponds to GD, and for $M_l \gg M_L = \Theta(1)$, the gradient approaches GD as well. We observed that both GN and BP get much closer to each other for sufficiently large widths. In other words, even when we realize GNT by setting $\bar{\gamma}_L = -1$, it has a quite close direction. A detailed view of the cosine similarity at the large width is shown in Figure 2(c). This result seems reasonable because in the context of second-order optimization, it has also been reported that GNT tends to collapse into an identity matrix owing to damping (Benzing, 2022). In summary, while PC's gradient switches between first-order GD and GNT depending on the parameterization, it is important to highlight that GNT behaves similarly to GD in the infinite-width limit.

As a minor extension, we can also analyze the nudge-type loss of PC defined by Eq. (S.10) (Alonso et al., 2022; Millidge et al., 2023; Pinchetti et al., 2024). In this case, the damping term $I$ in Eq. (15), is replaced by $(1 + \gamma/\beta)I$. Thus, the dependence on the parameterization remains essentially the same as that of the naive PC. Further discussion on nudge-type PC can be found in Appendix B.2.4.

### 4.3 STABILITY OF INFERENCE PHASE

To ensure feature learning in SGD, the $\mu$P framework requires stable activations, i.e., $\Delta u_{l<L} = \Theta(1)$. It seems natural to apply this requirement to the inference phase of PC. That is, let us suppose $u_{l<L,s}$

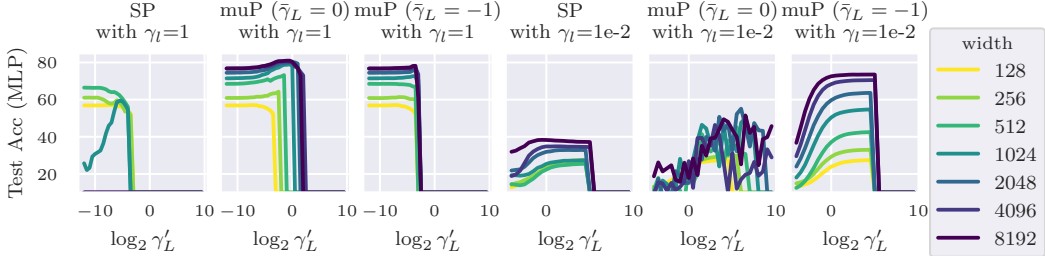

**Figure 5:** $\mu$**P with** $\bar{\gamma}_L = -1$ **performs consistently well, regardless of** $\gamma_l$**.** When $\gamma_l$ is small ($\gamma_l = 0.01$), $\mu$P with $\bar{\gamma}_L = 0$ performs poorly, while $\mu$P with $\bar{\gamma}_L = -1$ shows significantly better performance. This difference is likely due to slower inference convergence in $\mu$P with $\bar{\gamma}_L = 0$. For larger values of $\gamma_l$ ($\gamma_l = 1$), both $\mu$P configurations exhibit high accuracy. However, for $\mu$P with $\bar{\gamma}_L = 0$, $\gamma_L$ does not transfer effectively across widths, whereas $\mu$P with $\bar{\gamma}_L = -1$ demonstrates the successful transfer of $\gamma_L$ across widths.

varies by $\Theta(1)$ during the inference. Note that in Eq.(3) at $L-1$, the feedforward signal from the lower layer is $\gamma_{L-1} e_{L-1,s} = \Theta(1)$, and the error feedback from the last layer is $\gamma_L \phi'(u_{L-1,s}) \circ W_L^\top e_{L,s} = \Theta(1/M^{\bar{\gamma}_L + b_L})$. Both terms should be of order $\Theta(1)$ for the inference to successfully merge both feedforward and feedback signals. When $b_L = 1$, this condition requires $\gamma_L = \Theta(M)$, and we can expect the local loss in the last layer $e_L$ to decrease most prominently during the inference. Additionally for $\gamma_{l<L} = \Theta(1)$, the inference remains stable for layers $l < L - 1$. Empirical results in Figure 3 (left) confirm that when $\bar{\gamma}_L = -1$, the inference loss decreases consistently as the width increases, verifying that the "wider is better" hypothesis holds even in inference. This facilitates the hyperparameter transfer of $\gamma_L$ for the inference dynamics.

We also observe the benefits of using $\bar{\gamma}_L = -1$ for the parameter updates. Without F-ini, the convergence of inference usually deteriorates for SP, making inference stability especially critical in this scenario. As shown in Figure 3 (right), the "wider is better" trend holds with F-ini regardless of $\bar{\gamma}_L$. However, without F-ini, this trend holds only when $\bar{\gamma}_L = -1$. Figure 4 demonstrates that the $\mu$Transfer of the learning rate holds for $\bar{\gamma}_L = -1$. Additionally, Figure 5 indicates that $\bar{\gamma}_L = -1$ is also preferable from the perspective of $\mu$Transfer of $\gamma_L$.

## 5 FEATURE LEARNING OF TARGET PROPAGATION

### 5.1 $\mu$P OF TP

As overviewed in Section 3.1.2, TP reduces to GNT in the highly restrictive case of invertible networks. However, TP is not equivalent to GNT or BP in general cases (Meulemans et al., 2020; Ernoult et al., 2022). While TP involves two networks trained using different manners, and one may feel it challenging to obtain a stable parameterization for learning, we demonstrate that, under the assumption that the feedback network uses a linear activation function $\psi$, we can systematically derive $\mu$P for both TP and DTP.

**Theorem 5.1** ($\mu$P for TP and DTP (informal))**.** *Consider a linear feedback network. The forward network is allowed to have nonlinear activation functions. After the first training phase of $Q_l$, take the first one-step update of $W$. Then, we obtain $\mu$P as follows:*

$$\begin{cases} c_1 = 0, & c_{1<l<L} = 1, & c_L = 1, \\ b_1 = 0, & b_{1<l<L} = 1/2, & b_L = 1/2. \end{cases} \tag{17}$$

The derivation is presented in Section C.1. Note that the linear feedback network has trained weights in a pseudo-inverse form, that is, $Q_l^* = h_{l-1}(h_l^\top h_l + \mu I)^{-1} h_l^\top$. Stable parameterization can also be discussed for the training of the feedback network. For further details, see Section C.2.

As demonstrated in Figure 6, using the $\mu$P for TP results in the $\mu$Transfer appropriately across widths. Furthremore, Figure S.14 in Appendix tracks $\Delta h_l$ during training. In $\mu$P, $\Delta h_l$ remains consistent across different widths, whereas in SP, $\Delta h_l$ either diverges or diminishes as the width changes.

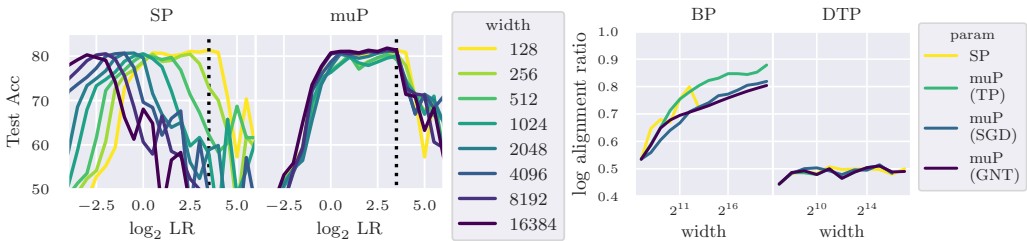

**Figure 6: (Left) $\mu$P can transfer the learning rate across widths in TP. (Right) TP does not have kernel regime.** We measured $\omega_L = \log_M(\|W_{L,0}\Delta h_{L-1,T}\|_{\text{RMS}}/\|W_{L,0}\|_{\text{RMS}}\|\Delta h_{L-1,T}\|_{\text{RMS}})$ across different parameterizations following Everett et al. (2024). In the infinite-width limit, $\omega_L$ converges to $\alpha$. Therefore, in TP, where $\omega_L$ remains fixed at 1/2 even as the width increases, the kernel regime disappears.

## 5.2 DISAPPEARANCE OF THE KERNEL REGIME

It is notable that $\mu$P in the previous work on the gradient methods requires $b_L = 1$; in TP, $\mu$P requires $b_L = 1/2$. For the usual gradient methods, a stable parameterization with $b_L = 1/2$ leads to the kernel regime. This raises the question: does a kernel regime exist in TP? Interestingly, in TP, the kernel regime disappears (see Corollary C.1 for the details).

*Rough sketch of derivation.* Condition 3.2 must hold to achieve stable learning in the hidden layers. Note that this condition is required in both the feature learning and kernel regimes. By expressing $\Delta h_{L-1} = \Theta(1/M^r)$ , we obtain

$$a_L + b_L + r - \alpha = 0. \tag{18}$$

When the inner product $W_L \Delta h_{L-1}$ follows the Law of Large Numbers (LLN), $\alpha = 1$, and when it follows the Central Limit Theorem, $\alpha = 1/2$ (Everett et al., 2024). Additionally, to prevent the output of the last layer from exploding, it is necessary that $h_L = O(1)$, that is, $a_L + b_L \geq 1/2$. Consequently, $r \leq \alpha - 1/2$. In BP, the dependence between $W_L$ and $\Delta h_{L-1}$ results in $\alpha = 1$ by the LLN. We have $r \leq 1/2$, allowing for the kernel regime. In contrast, in TP, updating the feedforward network weights does not induce a dependence between $W_L$ and $\Delta h_{L-1}$, leading to $\alpha = 1/2$. This is because the gradient is computed based on the feedback weight $Q_L^* = h_{L-1}(h_L^\top h_L + \mu I)^{-1}h_L^\top$, rather than $W_L$. Consequently, $r \leq 0$ and the kernel regime cannot be achieved in TP.

Figure 6 empirically confirms $\alpha = 1/2$ in TP. TP seems to be the first example in the infinite-width limit where $b_L = 1/2$ induces feature learning.

## 6 CONCLUSION

In this work, we revealed $\mu$P for local loss optimization that can effectively scale toward the infinite width in a stable manner, supported by our analysis of linear networks. Our study covers two of the most fundamental settings: the local targets computed during the inference phase (i.e., PC) and the feedback network (i.e., TP). Although neither method generally reduces to BP or GNT, making gradient computation non-trivial, we identified the $\mu$P and highlighted its intriguing properties, such as the gradient switching depending on the parameterization and the disappearance of the kernel regime. Additionally, we empirically confirmed that the derived $\mu$P facilitates hyperparameter transfer across widths.

**Limitation and future direction.** The derivation of $\mu$P assumes a one-step gradient and linear networks, although this prerequisite is not unique to our work (Yang & Hu, 2021; Yang et al., 2024). Ensuring the existence of feature learning dynamics for more general steps in the infinite width limit would require the development of a tensor program. However, handling the dependencies between variables that differ from standard BP, such as those arising from the inference phase and feedback pass, is non-trivial and presents an interesting direction for future research. Additionally, it would also be valuable to explore the learning dynamics of local learning and its convergence properties by extending the infinite width theory or further analyzing linear networks. We believe that understanding the universal behavior of large-scale limits will provide a foundation for the development of more effective algorithms.

ACKNOWLEDGMENTS

This work is supported by JST FOREST (Grant No. JPMJFR226Q) and JSPS KAEKENHI (Grant Nos. 22H05116, 23K16965). S.I. and R.Y. also acknowledge support from JST CREST Grant Number JPMJCR2112.

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

# Appendices

## A    EXTENDED BACKGROUND

### A.1    EXTENDED RELATED WORK

**Table S.1 :** List of Abbreviations

| Abbreviation | Full Name | Reference |
|---|---|---|
| BP | BackPropagation | - |
| SGD | Stochastic Gradient Descent | - |
| GNT | Gauss Newton Target | Proposition 3.3 |
| $\mu$P | Maximal Update Parameterization | Definition A.5 |
| NTK | Neural Tangent Kernel | - |
| HP | HyperParameter | - |
| PC | Predictive Coding | Section 3.1.1 |
| F-ini | Forward initialization | Technique i |
| FPA | Fixed prediction assumption | Technique ii |
| TP | Target Propagation | Section 3.1.2 |
| DTP | Difference Target Propagation | Eq S.13 |

### A.1.1    $\mu$P

While $\mu$P was introduced as a parameterization to induce a feature-learning regime in the infinite-width limit for theoretical interest (Yang & Hu, 2021), one practical advantage highlighted by Yang et al. (2021) is its ability to transfer the learning rate across different widths, a phenomenon they experimentally validated. This phenomenon, known as $\mu$Transfer, has also been examined from a theoretical perspective (Noci et al., 2024). Another notable advantage of $\mu$P is the improvement in the scaling law exponent, which has been investigated both experimentally (Qiu et al., 2024) and theoretically (Bordelon et al., 2024).

It is noteworthy that $\mu$P depends on the learning algorithm used and thus should be derived for each specific method. The $\mu$P for Adam was introduced in Yang et al. (2021), with its theoretical justification provided in Yang & Littwin (2023). For the second-order optimization, including the Gauss-Newton algorithm, K-FAC, and Shampoo, the $\mu$P was derived in Ishikawa & Karakida (2024). These works emphasize the importance of adjusting not only the learning rate but also the damping term in second-order optimization using $\mu$P. Additionally, $\mu$P was derived for Adafactor in Everett et al. (2024) and empirically demonstrated that the scaling of the $\epsilon$ term in Adam is also crucial in $\mu$P.

### A.1.2    PREDICTIVE CODING

Recent progress in deep learning has largely been achieved by the success of backpropagation (Rumelhart et al., 1986; LeCun et al., 1998; 2015). This success has increased the interest in exploring whether deep networks can also be trained using training algorithms other than backpropagation. This includes exploration into biologically plausible training methods and the benchmarking of local learning rules at modern-scale networks on deep-learning benchmark datasets; equilibrium propagation (Scellier & Bengio, 2017; Laborieux et al., 2021), target propagation (Bengio, 2014; Lee et al., 2015), predictive coding (Whittington & Bogacz, 2017; Song et al., 2020; Salvatori et al., 2023), and forward-forward algorithms (Ren et al., 2023).

**Computation of predictive coding for supervised learning:** Predictive coding was originally introduced as an algorithm for solving inverse problems where the goal is to find the parameters $W$ that maximize the marginal likelihood $p(v_L; W)$ where $v_L$ denotes a variable representing the

network output and $v_L = x$ for the input data $x$ (Rao & Ballard, 1999). In this inverse problem, we consider latent variables $v_l$ (also referred to as causes) and a generative function $p(v_l|v_{l-1}; W_l)$. If we assume a hierarchical Gaussian generative model, the marginal probability of the causes is as follows:

$$p(v_0, \ldots, v_L; W) = p(v_0) \prod_{l=1}^{L} p(v_l|v_{l-1}; W_l) \tag{S.1}$$

$$= \mathcal{N}\left(v_0, \gamma_0^{-1}I\right) \prod_{l=1}^{L} \mathcal{N}\left(W_l\phi(v_{l-1}), \gamma_l^{-1}I\right). \tag{S.2}$$

Friston (2003) framed this inverse problem as an EM algorithm aimed at minimizing the following variational free energy:

$$\mathcal{F} = \mathrm{KL}(q(v_0, \ldots, v_L) \| p(v_0, \ldots, v_L; W)), \tag{S.3}$$

where $q(v_0, \ldots, v_L)$ is a tractable posterior probability distribution for the EM algorithm. In particular, the E-step, referred to as inference, minimizes variational free energy by causes $v_l$, while the M-step, referred to as learning, minimizes by parameters $W_l$. We usually apply a mean-field approximation or a Laplace approximation to the tractable probability distribution (Friston, 2005; Salvatori et al., 2024a;b). Under these formulations, we can derive the variational free energy for PC.

$$\mathcal{F} = \sum_{l=1}^{L} \gamma_l \frac{1}{2} \|v_l - W_l\phi(v_{l-1})\|^2. \tag{S.4}$$

While predictive coding networks were originally discussed primarily in the context of generative models for unsupervised learning, Whittington & Bogacz (2017) reformulated PC networks for supervised learning and highlighted their potential for use in the context of deep learning. Specifically, if we fix $v_0 = x$ and $v_L = y$ for data, this corresponds to supervised learning using the mean squared loss.

**Heuristic techniques in PC:** After Whittington & Bogacz (2017), there has been an increasing amount of studies evaluating the performance of PC networks as local learning on deep-learning benchmark datasets (Salvatori et al., 2023; Pinchetti et al., 2024). They revealed that the original implementation of PC networks is insufficient for achieving stable training performance, and heuristic modifications have played an important role. For instance, Fixed prediction assumption (FPA) has been introduced to achieve higher performance by approximating the gradient computation of PC networks closer to that of BP (Millidge et al., 2022b; Rosenbaum, 2022). Under FPA, $\phi'(v_{l,s})$ is replaced with $\phi'(v_{l,0})$, resulting in an inference phase given by

$$v_{l,s+1} = v_{l,s} - \gamma_l e_{l,s} + \gamma_{l+1}\phi'(v_{l,0}) \circ W_{l+1}^\top e_{l+1,s}, \quad (l < L) \tag{S.5}$$

and a learning phase is given by

$$W_{l,t+1} = W_{l,t} + \eta_l \gamma_l e_{l,s}\phi(v_{l-1,0})^\top. \tag{S.6}$$

Additionally, FPA is typically used with Forward initialization (F-ini). In Forward initialization, the state $v_{l,0}$ is initialized with the forward value $u_{l,0}$. While F-ini has been implicitly utilized in most studies on PC (Whittington & Bogacz, 2017; Song et al., 2020; Rosenbaum, 2022), its role was explicitly highlighted in Alonso et al. (2022), where the authors compared the convergence with and without F-ini.

Rosenbaum (2022) pointed out that when both F-ini and FPA are assumed, PC networks are entirely reduced to BP and that if the algorithms are fully equivalent to BP, the advantages of biological plausibility and local updates are lost. In this study, we aim to identify a parameterization that enables stable local learning while maintaining distinctions from BP. By leveraging several recently developed heuristics, we clarify the desirable scales for stable and efficient local learning.

**Nudged PC:** The design of loss functions in PC networks has also been a focus of algorithmic improvements. In most of the ML research, classification tasks generally use cross-entropy loss rather than mean squared loss. Accordingly, PC networks sometimes use cross-entropy loss as well (Pinchetti et al., 2024). The free energy for a general loss function is given by

$$\mathcal{F} = \mathcal{L}(y, W_L\phi(v_{L-1})) + \sum_{l=1}^{L-1} \gamma_l \frac{1}{2} \|v_l - W_l\phi(v_{l-1})\|^2. \tag{S.7}$$

---

**Algorithm 1** PC Algorithm (Simultaneous or Sequential inference)

---

1: **for** $s = 1$ to $n$ **do**
2:      $e_{L,s} \leftarrow \nabla_{u_{L,s}} \mathcal{L}(W_L \phi_L(u_{L-1,s}), y)$
3:      **for** $l = L - 1$ to 1 **do**
4:          $e_{l,s} \leftarrow u_{l,s} - W_l \phi_{l-1}(u_{l-1,s})$
5:          $e_{l+1,s} \leftarrow u_{l+1,s+1} - W_{l+1} \phi_l(u_{l,s})$ **(Sequential Inference)**
6:          $u_{l,s+1} \leftarrow u_{l,s} - \gamma_l e_{l,s} + \gamma_{l+1} \phi'(u_{l,s}) \circ W_{l+1}^\top e_{l+1,s}$
7:      **end for**
8: **end for**

---

and its update rule for the inference phase is given by

$$v_{l,s+1} = v_{l,s} - \gamma_l e_{l,s} + \gamma_{l+1} \phi'(v_{l,s}) \circ W_{l+1}^\top e_{l+1,s}, \quad (l < L) \tag{S.8}$$

$$v_{L,s+1} = v_{L,s} - \gamma_L \frac{\partial \mathcal{L}}{\partial v_L}. \tag{S.9}$$

Furthermore, there are formulations of PC networks that incorporate the nudge term introduced in equilibrium propagation (Scellier & Bengio, 2017). PC networks with a nudge term update the state $v_l$ and weights $W_l$ to minimize the following free energy function (Alonso et al., 2022; Millidge et al., 2023; Pinchetti et al., 2024):

$$\mathcal{F} = \beta \mathcal{L}(y, v_L) + \sum_{l=1}^{L} \gamma_l \frac{1}{2} \| v_l - W_l \phi(v_{l-1}) \|^2 . \tag{S.10}$$

Here, $\beta$ is a nudge coefficient parameter that covers some variants of the PC algorithms in the previous work.

There are several possible orders for computing this inference (Alonso et al., 2024) as illustrated in Algorithm 1. Specifically, $e_l$ can be calculated sequentially from the output layer to the input layer, or it can be updated synchronously across all layers simultaneously from the output to the input. While the main text focuses on Sequential Inference (SI), where computations proceed layer-by-layer from the output to the input, Predictive Coding with synchronous inference is also a valid approach worth considering.

### A.1.3 TARGET PROPAGATION

Target propagation offers a learning rule that is more biologically plausible and easier for the brain to implement compared to BP (Bengio, 2014). Specifically, it addresses the following two issues inherent to BP (Meulemans et al., 2020).

1. **Signed error transmission problem:** BP propagates the error gradient to the lower layers, whereas the brain propagates target values for the neurons (Lillicrap et al., 2020).
2. **Weight transport problem:** BP requires exact weight symmetry between the forward and backward paths. However, the brain cannot transport weights (Grossberg, 1987; Akrout et al., 2019).

Target propagation aims to address the two issues by:

1. Propagating the target value $\hat{h}_L = h_L - \hat{\eta} \nabla_{h_L} \mathcal{L}$ instead of the error gradient $\nabla_{h_L} \mathcal{L}$ for signed error transmission problem.
2. Utilizing a feedback network $g_l$ distinct from the feedforward network $f_l$ to propagate the target value $\hat{h}_L$ for weight transport problem.

The feedback network $g_l$ has the weights $Q_l$ distinct from those in the feedforward network.

$$\hat{h}_l = g_l(\hat{h}_{l+1}), \quad g_l(x) = \psi(Q_l x) \quad (l = 1, ..., L - 1). \tag{S.11}$$

Here, $\psi$ denotes the activation function. While it is often the same as the activation function used in the feedforward network, it is also possible to consider a different activation function. We set $\psi = \phi$

in experiments and assume $\psi$ as the identity function only in Theorem 5.1. The feedback network is trained to minimize the following reconstruction loss:

$$\mathcal{L}_{\mathbf{rec}}(Q_l) = \|g_l \left(f_l(h_{l-1} + \epsilon)\right) - h_{l-1} + \epsilon\|^2, \tag{S.12}$$

where $\epsilon$ is a small Gaussian noise to improve the robustness of the feedback network. Target propagation attempts to approximate the inverse function of the feedforward network by learning the feedback network through the optimization of this reconstruction loss.

Difference target propagation (DTP) is an improved method of TP, which adjusts the propagation in the feedback network as follows (Lee et al., 2015).

$$\hat{h}_i = g_i^{\mathrm{diff}} \left(\hat{h}_{i+1}, h_{i+1}, h_i\right) = g_i\left(\hat{h}_{i+1}\right) - \left(g_i\left(h_{i+1}\right) - h_i\right). \tag{S.13}$$

In TP, the accumulation of the reconstruction error $g_i(f_i(h_i)) - h_i$ during propagation poses an obstacle to optimization. In DTP, subtracting $\left(g_i\left(h_{i+1}\right) - h_i\right)$ mitigates the accumulation of the reconstruction error and improves the progress of learning.

As a side note, Meulemans et al. (2020) and Bengio (2020) pointed out that TP can be related to the Gauss-Newton method for invertible networks. Additionally, Meulemans et al. (2020) proposed Direct Difference Target Propagation so as to establish this correspondence even in non-invertible networks under some infinitesimal conditions. Ernoult et al. (2022) reported that one can stabilize TP by introducing the additional Local Difference Reconstruction Loss which makes the gradient align more closely with Backpropagation rather than Gauss-Newton Targets. In our work, we aim to clarify the fundamental properties of TP and DTP from the perspective of parameterization and do not consider such additional conditions or loss functions.

## A.2 DEFINITIONS FOR STABLE PARAMETERIZATION

As is common in the $\mu$P theory, we also assume that the firing activities are of order 1 at random initialization:

**Assumption A.1.** $u_{l,0}, h_{l,0} = \Theta(1)$ $(l < L)$, $\quad f_0 = u_{L,0} = O(1)$.

As shown in Theorem H.6 of Yang & Hu (2021), this assumption immediately leads to

$$a_1 + b_1 = 0, \quad a_{1<l<L} + b_{1<l<L} = 1/2, \quad a_L + b_L \geq 1/2. \tag{S.14}$$

In addition, the stability of learning is defined as follows (see Definition H.4 in Yang & Hu (2021) for more detail):

**Definition A.2** (Stability of learning). We say an abc-parameterization is stable if, for $l < L$ and for any fixed $t \geq 1$,

$$\Delta h_{l,t} = O(1), \ \Delta f_t = O(1), \tag{S.15}$$

under Assumption A.1.

Condition S.15 ensures avoiding exploding dynamics with respect to the width, i.e., $\Delta h_{l,t} = O(1/M^k)$ with $k < 0$.

We follow the derivation based on the infinitesimal one-step update from random initialization (Yang & Hu, 2021; Ishikawa & Karakida, 2024), which involves taking the limit of a sufficiently small coefficient of the learning rate $\eta'$. This formulation clarifies the proof and enables the systematic derivation of $\mu$P across various problems. In the infinitesimal formulation, Conditions 3.1 and 3.2 are expressed as follows:

**Condition A.3** ($W_l$ updated maximally).

$$\partial_{\eta'} \Delta W_l h_{l-1,1}\big|_{\eta'=0} = \Theta(1) \tag{S.16}$$

where $\Delta W_l := W_{l,1} - W_{l,0}$.

**Condition A.4** ($W_L$ initialized maximally).

$$\partial_{\eta'} W_{L,0} \Delta u_{L-1,1}\big|_{\eta'=0} = \Theta(1). \tag{S.17}$$

As described in the previous work (Yang & Hu, 2021; Ishikawa & Karakida, 2024), Condition 3.1 (or A.3) naturally appears from the expansion of Eq. (10) by the parameter update, yielding

$$\partial_{\eta'} \Delta W_l h_{l-1,1} \big|_{\eta'=0} = \Theta(1/M^{r_l}), \tag{S.18}$$

for abc-parameterization. The stability requires

$$r_l \geq 0 \tag{S.19}$$

and, in particular, feature learning is characterized by $r_l = 0$. Condition 3.2 (or A.4) is required to eliminate an uninteresting case in which the hidden layer provides no contribution to the network output. Both NTK and feature learning regimes are characterized by this condition.

As is shown in Yang & Hu (2021), $\mu$P is the unique stable parameterization satisfying Condition A.3 for $l \leq L$ and Condition A.4 for $W_L$. Thus, we can admit this characterization as a definition of $\mu$P.

**Definition A.5** ($\mu$P). $\mu$P is the stable abc parameterization satisfying Condition A.3 for $l \leq L$ and Condition A.4 for $W_L$.

Note that Condition A.3 is required not only for hidden layers but also for the last layer. In the previous work, this eliminates a *trivial* case of learning, i.e., $\Delta h_{L,t} = O(1/M^k)$ with $k > 0$, where the effect of learning vanishes.

# B $\mu$P OF PREDICTIVE CODING

## B.1 DERIVATION FOR PREDICTIVE CODING WITH SINGLE-SHOT SI

**Theorem B.1** (Stable parameterization for PC). *Set inference step sizes $\gamma_{l<L} = \Theta(1)$ and $\gamma_L = \gamma'/M^{\bar{\gamma}_L}$ with a positive constant $\gamma'$. Suppose F-ini and single-shot sequential inference, and consider a one-step update of parameters after the inference. For infinitesimal step sizes $\gamma'_L$ and $\eta'$, PC admits the $\mu$P for feature learning at*

$$\begin{cases} c_1 = -\bar{\gamma}_L - 1, & c_{1<l<L} = -\bar{\gamma}_L, & c_{l=L} = 1, \\ b_1 = 0, & b_{l<L} = 1/2, & b_L = 1. \end{cases} \tag{S.20}$$

*Additionally, it admits the NTK parameterization at*

$$\begin{cases} c_1 = -\bar{\gamma}_L, & c_{1<l<L} = 1 - \bar{\gamma}_L, & c_L = 1 \\ b_1 = 0, & b_{l<L} = 1/2. \end{cases} \tag{S.21}$$

*Proof.* Assuming F-ini, considering the single-shot SI for $v_l$, we have

$$v_{l,1} = v_{l,0} + \gamma_{l+1} \phi'(v_{l,0}) \circ W_{l+1}^\top e_{l+1,1} \tag{S.22}$$

$$= v_{l,0} + \gamma_{l+1} \phi'(v_{l,0}) \circ W_{l+1}^\top (v_{l+1,1} - W_{l+1} h_{l,0}) \tag{S.23}$$

$$= v_{l,0} + \gamma_{l+1} \phi'(v_{l,0}) \circ W_{l+1}^\top (v_{l+1,1} - v_{l+1,0}), \tag{S.24}$$

for $l < L$ where $e_{L,0} = y - W_L v_{L-1,0} =: \delta_L$. When the CE loss is used instead of MSE loss, $\delta_L = y - f$ becomes $\delta_L = y - \text{softmax}(f)$, and the order analysis remains unchanged. To keep the notation concise, we set $M_L = 1$ in this proof. A generalization for $M_L = \Theta(1)$ is possible. Next, we define

$$\delta_{l<L} := \partial u_L / \partial u_l. \tag{S.25}$$

Note that a batch gradient can be used with $N$ training samples where $N = O(1)$. One can regard $v_l$ as an $M \times N$ matrix in the derivation.

Using

$$v_{l,1} - v_{l,0} = - \prod_{i=l+1}^{L} \gamma_i \delta_l, \tag{S.26}$$

we have

$$e_{l,1} := v_{l,1} - \phi(W_l v_{l-1,1}) \tag{S.27}$$

$$= (u_{l,0} - \prod_{i=l+1}^{L} \gamma_i \delta_l) - \phi(W_l (u_{l-1,0} - \prod_{i=l}^{L} \gamma_i \delta_{l-1})) \tag{S.28}$$

for $l = 1, ..., L - 1$. Recall that $v_{l,0} = u_{l,0}$ for F-ini. For $l = L$,

$$e_{L,1} := y - W_L h_{L-1,1} \tag{S.29}$$

$$= y - W_L \phi(u_{L-1,0} - \gamma_L \delta_{L-1} \mathrm{diag}(\delta_L)) \tag{S.30}$$

where $\mathrm{diag}(x)$ denotes a diagonal matrix whose diagonal entries are given by $x$. The above equation comes from

$$v_{L-1,1} = u_{L-1,0} - \frac{\gamma_L}{2} \nabla_{v_{L-1}} \| y - W_L h_{L-1} \|^2 \tag{S.31}$$

$$= u_{L-1,0} - \gamma_L \phi'_{L-1} \circ W_L^\top \delta_L = u_{L-1,0} - \gamma_L \delta_{L-1} \mathrm{diag}(\delta_L). \tag{S.32}$$

The first one-step update of the weight is expressed as

$$\Delta W_{l,1} = \frac{\eta'}{M^{2a_l + c_l}} e_{l,1} h_{l-1,1}^\top, \tag{S.33}$$

In PC, in addition to the usual learning rate $\eta$, there also exists $\gamma$. Therefore, in addition to the infinitesimal update of the learning rate $\eta$ for the weight update, we also consider the infinitesimal inference step size $\gamma_L$. By applying the perturbation of $\gamma_L$ to Conditions A.3 and A.4, we derive

$$\Delta U_l + \partial_{\gamma'_L} \Delta U_l \big|_{\gamma'_L = 0} \gamma'_L = \Theta(1), \tag{S.34}$$

$$\Delta V_L + \partial_{\gamma'_L} \Delta V_L \big|_{\gamma'_L = 0} \gamma'_L = \Theta(1) \tag{S.35}$$

where we define

$$\Delta U_l := \partial_{\eta'} \Delta W_{l,1} h_{l-1,s=1} \big|_{\eta'=0}, \tag{S.36}$$

$$\Delta V_L := \partial_{\eta'} W_{L,0} \Delta h_{l-1,s=1} \big|_{\eta'=0}. \tag{S.37}$$

It is noteworthy that we retain the zero-th order terms, namely, $\Delta U_l$ and $\Delta V_l$ in the conditions. This is because, even without the inference phase, parameter updates can progress while the internal states remain at their initialization. Therefore, even if the maximalization of the order is less than $\Theta(1)$ in the first-order perturbation terms, stable learning can still occur. Since $\mu$P aims to maximize the order of updates as much as possible, we require the first-order terms of Eqs. (S.34,S.35) to be $\Theta(1)$ whenever possible.

We introduce the following kernel matrix:

$$K_l^A := h_l^\top h_l / M. \tag{S.38}$$

For the random initialization $W_l$, from Eq. (S.14), we asymptotically obtain

$$K_l^A = \Theta(1), \quad K_L^A = \Theta(1/M^{2(a_L + b_L)}) \tag{S.39}$$

in the infinite-width limit (Yang, 2020).

**On Condition A.1.**

**(i) Case of $1 < l < L$.**

$$\partial_{\gamma'_L} \Delta U_l \big|_{\gamma'_L = 0} = \partial \left( \frac{1}{M^{\theta_l}} e_{l,s=1} h_{l-1,s=1}^\top h_{l-1,s=1} \right) \Big|_{\gamma'_L = 0} \tag{S.40}$$

$$= \frac{1}{M^{\theta_l}} \partial(e_{l,s=1}) h_{l-1,s=1}^\top h_{l-1,s=1} \big|_{\gamma'_L = 0} + \frac{1}{M^{\theta_l}} e_{l,s=1} \partial(h_{l-1,s=1}^\top h_{l-1,s=1}) \big|_{\gamma'_L = 0}$$

$$= \frac{1}{M^{\theta_l}} \partial(e_{l,s=1}) \big|_{\gamma'_L = 0} h_{l-1,s=0}^\top h_{l-1,s=0} \tag{S.41}$$

$$= \frac{1}{M^{\theta_l + \bar{\gamma}_L - 1}} \left( -\delta_l + \phi'(W_l u_{l-1,0}) \circ W_l \delta_{l-1} \right) K_{l-1}^A \tag{S.42}$$

where we used $e_{l<L,s=1} \big|_{\gamma'_L = 0} = 0$ and $h_{l,s=1} \big|_{\gamma'_L = 0} = h_{l,s=0}$. Since $\delta_{l<L} = \Theta(W_L) = \Theta(1/M^{a_L + b_L})$ and $\Delta U_l \sim M^{\theta_l + a_L + b_L - 1}$, we have

$$\Delta U_l + \partial_{\gamma'_L} \Delta U_l \big|_{\gamma'_L = 0} \gamma'_L \sim 1/M^{\min\{\theta_l + a_L + b_L - 1, \theta_l + \bar{\gamma}_L + a_L + b_L - 1\}}. \tag{S.43}$$

The similarity symbol ("$\sim$") denotes that the left-hand side is of the same order as the right-hand side. Note that if the first-order term becomes negligible, the contribution of the inference phase disappears in the parameter update. To maximize the order of the first-order term, we require

$$\bar{\gamma}_L \leq 0 \tag{S.44}$$

and obtain

$$r_l = \theta_l + \bar{\gamma}_L + a_L + b_L - 1. \tag{S.45}$$

**(ii) Case of $l = 1$.**

$$\partial_{\gamma'_L} \Delta U_l \big|_{\gamma'_L=0} = -\frac{1}{M^{\theta_l+\bar{\gamma}_L}} \delta_l K_0^A. \tag{S.46}$$

$$\sim 1/M^{\theta_1+\bar{\gamma}_L+a_L+b_L} \tag{S.47}$$

Here, we used $e_{l<L,s=1}\big|_{\gamma'_L=0} = 0$ and $h_{l,s=1}\big|_{\gamma'_L=0} = h_{l,s=0}$. Similar to the case of $1 < l < L$, Condition (S.34) leads to $\bar{\gamma} \leq 0$ and

$$r_l = \theta_1 + \bar{\gamma}_L + a_L + b_L. \tag{S.48}$$

**(iii) Case of $l = L$.**

$$\partial_{\gamma'_L} \Delta U_L \big|_{\gamma'_L=0} \tag{S.49}$$

$$= \frac{1}{M^{\theta_L}} \partial(e_{L,s=1}) h_{L-1,s=1}^\top h_{L-1,s=1}\big|_{\gamma'_L=0} + \frac{1}{M^{\theta_L}} e_{L,s=1} \partial(h_{L-1,s=1}^\top h_{L-1,s=1})\big|_{\gamma'_L=0} \tag{S.50}$$

$$= \frac{1}{M^{\theta_L}} \partial(e_{L,s=1})\big|_{\gamma'_L=0} h_{L-1,s=0}^\top h_{L-1,s=0} + \frac{1}{M^{\theta_L}} e_{L,s=1} \partial(h_{L-1,s=1}^\top h_{L-1,s=1})\big|_{\gamma'_L=0} \tag{S.51}$$

$$= \frac{1}{M^{\theta_L+\bar{\gamma}_L-1}} \phi'(W_L u_{L-1,0}) \circ W_L \delta_{L-1} K_{L-1}^A + \frac{1}{M^{\theta_L}} \delta_L \partial(h_{L-1,s=1}^\top h_{L-1,s=1})\big|_{\gamma'_L=0} \tag{S.52}$$

$$= \frac{1}{M^{\theta_L+\bar{\gamma}_L-1}} \phi'(W_L u_{L-1,0}) \circ W_L \delta_{L-1} K_{L-1}^A + \frac{2}{M^{\theta_L}} \delta_L h_{L-1,s=1}^\top \partial(h_{L-1,s=1})\big|_{\gamma'_L=0} \tag{S.53}$$

Note that from Eq. (S.30), we have

$$e_{L,s=1}\big|_{\gamma'_L=0} = -(W_L(\phi'_{L-1} \circ \delta_{L-1})) \circ \delta_L / M^{\bar{\gamma}_L}. \tag{S.54}$$

Since $e_{L,s=1}\big|_{\gamma'_L=0} = \delta_L \neq 0$,

$$W_L(\phi'_{L-1} \circ \delta_{L-1})) \circ \delta_L = \Theta(1/M^{2(a_L+b_L)-1}). \tag{S.55}$$

For the second term in Eq. (S.53), we have

$$h_{L-1,s=1}^\top \partial(h_{L-1,s=1})\big|_{\gamma'_L=0} = h_{L-1,s=0}^\top \partial\phi(W_{L-1} v_{L-2,1})\big|_{\gamma'_L=0} \tag{S.56}$$

$$= h_{L-1}^\top (\phi'_{L-1} \circ W_{L-1} \partial v_{L-2,1})\big|_{\gamma'_L=0} \tag{S.57}$$

$$= -\frac{\gamma_{L-1}}{M^{\bar{\gamma}_L}} h_{L-1}^\top (\phi'_{L-1} \circ W_{L-1} \delta_{L-2}) \tag{S.58}$$

where we used $v_{L-2,1} - v_{L-2,0} = -\gamma_{L-1}\gamma_L \delta_{L-2}$ from Eq. (S.26). Let us recall that a variable without an index indicates the initial state at $s = 0$. The variable $\delta_{L-1}$ includes $W_L$ whereas $h_{L-1}$ is independent of it. Therefore, by applying the Central Limit Theorem with respect to $W_L$, we have

$$h_{L-1,s=1}^\top \partial(h_{L-1,s=1})\big|_{\gamma'_L=0} \sim 1/M^{\bar{\gamma}_L+a_L+b_L-1/2}. \tag{S.59}$$

Then,

$$\partial_{\gamma'_L} \Delta U_L\big|_{\gamma'=0} \sim 1/M^{\min\{\theta_L+\bar{\gamma}_L+2(a_L+b_L)-2,\theta_L+\bar{\gamma}_L+a_L+b_L-1/2\}}. \tag{S.60}$$

In contrast, we have

$$\Delta U_L \sim 1/M^{\theta_L-1}. \tag{S.61}$$

Comparing the zero-th and first order terms (S.60,S.61), we obtain

$$\min\{\theta_L - 1, \theta_L + \bar{\gamma}_L + 2(a_L + b_L) - 2, \theta_L + \bar{\gamma}_L + a_L + b_L - 1/2\} = 0 \quad \text{(S.62)}$$

Because $a_L + b_L - 1/2 \geq 0$ from Eq. (S.14), we obtain

$$\theta_L - 1 = 0. \quad \text{(S.63)}$$

**On Condition A.2.**

$$\partial_{\gamma'_L} \Delta V_L \big|_{\gamma'_L = 0} = W_{L,0}(\phi'(u_{L-1}) \circ \partial_{\gamma'_L} \partial_{\eta'}(\Delta W_{L-1,1} h_{L-2,1}) \big|_{\eta'=0, \gamma'_L=0})$$

$$= e_M(\delta_{L-1} \circ \frac{1}{M^{\theta_{L-1}}} \partial_{\gamma'}(\Delta W_{L-1} h_{L-2}) \big|_{\gamma'_L=0}) \quad \text{(S.64)}$$

$$= \Theta(1/M^{a_L + b_L + r_{L-1} - 1}) \quad \text{(S.65)}$$

where $e_M$ denotes an $M$-dimensional vector with all entries equal to 1. Note that the product with $e_M$ means the summation over $M$.

Finally, from Conditions 1 and 2, the $\mu$P is given by

$$\theta_1 + \bar{\gamma}_L + a_L + b_L = 0 \quad (l = 1), \quad \text{(S.66)}$$
$$\theta_l + \bar{\gamma}_L + a_L + b_L - 1 = 0 \quad (1 < l < L), \quad \text{(S.67)}$$
$$\theta_L - 1 = 0 \quad (l = L) \quad \text{(S.68)}$$
$$a_L + b_L - 1 = 0, \quad \text{(S.69)}$$

and $\bar{\gamma}_L \leq 0$. That is,

$$\begin{cases} c_1 = -\bar{\gamma}_L - 1, \quad c_{1 < l < L} = -\bar{\gamma}_L \geq 0, \quad c_{l=L} = 1, \\ b_1 = 0, \quad b_{l<L} = 1/2, \quad b_L = 1. \end{cases} \quad \text{(S.70)}$$

The above $\mu$P case assumes $a_l = 0$. It is important to note that there is no issue in replacing $c_l$ with $\theta_l = 2a_l + c_l$, which introduces an indeterminacy of $a_l = a_l + \alpha$ and $c_l = c_l - 2\alpha$.

We can also derive the NTK parameterization, which is a commonly used term for the kernel regime for $r_{l<L} = 1/2$ (Yang & Hu, 2021):

$$\theta_1 + \bar{\gamma}_L + a_L + b_L = 1/2 \quad (l = 1), \quad \text{(S.71)}$$
$$\theta_l + \bar{\gamma}_L + a_L + b_L - 1 = 1/2 \quad (1 < l < L), \quad \text{(S.72)}$$
$$\theta_L - 1 = 0, \quad (l = L) \quad \text{(S.73)}$$
$$a_L + b_L - 1/2 = 0. \quad \text{(S.74)}$$

$$\square$$

It is noteworthy that the gradient computed by Eq. (S.28) differs from $\delta_l$ in standard SGD, implying that the NTK matrix also deviates from $\nabla_\theta f^\top \nabla_\theta f$. Even in this case, the NTK regime can emerge with a certain modified kernel composed of $e_l$ and $h_l$. A similar situation arises in the NTK regime of second-order optimization (Karakida & Osawa, 2020). Although the preconditioner modifies the NTK matrix, the linearization of the model still holds, allowing the emergence of the kernel regime.

## B.2 FIXED POINTS OF PC IN LINEAR NETWORKS

### B.2.1 PROOF FOR THEOREM 4.2

In this section, we analyze the fixed point of the inference phase using a linear network:

$$f(x) = W_l W_{L-1} ... W_1 x. \quad \text{(S.75)}$$

Even for linear networks, the properties of the fixed points have rarely been analyzed. An exception is a recent study by Innocenti et al. (2024). They explicitly derived the free energy at a fixed point to analyze the parameter loss landscape of a naive PC. However, their analysis uses an unfolding calculation of a hierarchical Gaussian model to directly derive the free energy. Although this is an elegant derivation, it is not a method for explicitly obtaining the fixed points themself. Additionally,

since their proof is based on $\gamma = 1$, we need another method to determine the dependence on the inference size. Here, we provide a derivation of the states at the fixed point that can be used more generally for various inference sizes and add a nudge term (in Section B.2.4).

*Proof*. We consider the inference of naive PC:

$$F(v_1, ..., v_L) = \frac{\gamma_L}{2} \|y - W_L v_{L-1}\|^2 + \sum_{l=1}^{L-1} \frac{\gamma_l}{2} \|v_l - W_l v_{l-1}\|^2. \tag{S.76}$$

Taking $\frac{\partial F}{\partial v_l} = 0$, we obtain the following fixed-point equations:

$$-\gamma_L W_L^\top (y - W_L v_{L-1}) + \gamma_{L-1}(v_{L-1} - W_{L-1} v_{L-2}) = 0, \quad (l = L) \tag{S.77}$$

$$-\gamma_l W_l^\top (v_l - W_l v_{l-1}) + \gamma_{l-1}(v_{l-1} - W_{l-1} v_{l-2}) = 0, \quad (1 < l < L) \tag{S.78}$$

$$-\gamma_2 W_2^\top (v_2 - W_2 v_1) + \gamma_1(v_1 - W_1 x) = 0 \quad (l = 1). \tag{S.79}$$

These equations are summarized in the following matrix form:

$$\begin{bmatrix} I & O & \cdots & \cdots & O \\ -W_L^\top & I & O & \cdots & O \\ O & -W_{L-1}^\top & I & \cdots & O \\ \vdots & \ddots & \ddots & & \vdots \\ O & \cdots & O & -W_2^\top & I \end{bmatrix} \begin{bmatrix} \gamma_{L-1} e_{L-1}^* \\ \gamma_{L-2} e_{L-2}^* \\ \vdots \\ \gamma_2 e_2^* \\ \gamma_1 e_1^* \end{bmatrix} = \begin{bmatrix} \gamma_L W_L^\top e_L^* \\ O \\ O \\ \vdots \\ O \end{bmatrix} \tag{S.80}$$

where $e_l^* := v_l^* - W_l v_{l-1}^*$ and $e_L^* := y - W_L v_{L-1}^*$.

Here, we use the following lemma:

**Lemma B.2.** *Define*

$$A_L := \begin{bmatrix} I & O & \cdots & \cdots & O \\ -W_L^\top & I & O & \cdots & O \\ O & -W_{L-1}^\top & I & \cdots & O \\ \vdots & \ddots & \ddots & & \vdots \\ O & \cdots & O & -W_2^\top & I \end{bmatrix}. \tag{S.81}$$

*Its inverse matrix is given by*

$$A_L^{-1} = \begin{bmatrix} I & O & \cdots & \cdots & O \\ W_L^\top & I & O & \cdots & O \\ W_{L-1:L}^\top & W_{L-1}^\top & I & \cdots & O \\ \vdots & \ddots & \ddots & & \vdots \\ W_{2:L}^\top & \cdots & W_{2:3}^\top & W_2^\top & I \end{bmatrix}. \tag{S.82}$$

*Proof.* One can easily derive this inverse matrix. A simple derivation is achieved by induction. We can express

$$A_L = \begin{bmatrix} I & O \\ K & A_{L-1} \end{bmatrix} \tag{S.83}$$

where $K^\top = [W_L, O, ..., O]$. Suppose that the inverse of $A_{L-1}$ is given by Eq. (S.82). Then,

$$A_L^{-1} = \begin{bmatrix} I & O \\ K A_{L-1}^{-1} & A_{L-1}^{-1} \end{bmatrix}. \tag{S.84}$$

Since $K A_{L-1}^{-1} = [W_L, W_{L-1:L}, ..., W_{2:L}]^\top$, the inversion of $A_L$ is also given by Eq. (S.82).

$\square$

By using Lemma B.2, we can transform Eq. (S.80) as follows:

$$\begin{bmatrix} e_{L-1}^* \\ e_{L-2}^* \\ \vdots \\ e_2^* \\ e_1^* \end{bmatrix} = \gamma_L \begin{bmatrix} \frac{1}{\gamma_{L-1}} W_L^\top e_L^* \\ \frac{1}{\gamma_{L-2}} W_{L-1:L}^\top e_L^* \\ \vdots \\ \frac{1}{\gamma_2} W_{2:L}^\top e_L^* \end{bmatrix}. \tag{S.85}$$

Although this equation can not be solved explicitly for $v_{l-1}^*$, we can, nonetheless, solve it by summing over $e_l^*$ as follows:

$$v_{L-1}^* - W_{L-1:1}x \tag{S.86}$$

$$= e_{L-1}^* + W_{L-1}e_{L-2}^* + \cdots + W_{L-1:2}e_1^*$$

$$= \gamma_L \left( \frac{1}{\gamma_{L-1}}I + \frac{1}{\gamma_{L-2}}W_{L-1}W_{L-1}^\top + \cdots + \frac{1}{\gamma_1}W_{L-1:2}W_{L-1:2}^\top \right)W_L^\top e_L^*. \tag{S.87}$$

This leads to

$$v_{L-1}^* = \left( I + \frac{\gamma_L}{\gamma_{L-1}}W_L^\top W_L + \frac{\gamma_L}{\gamma_{L-2}}W_{L-1}W_{L-1}^\top W_L^\top W_L + \cdots + \frac{\gamma_L}{\gamma_1}W_{L-1:2}W_{L-1:2}^\top W_L^\top W_L \right)^{-1}$$

$$\cdot \left( W_{L-1:1}x + \gamma_L \left( \frac{1}{\gamma_{L-1}}I + \frac{1}{\gamma_{L-2}}W_{L-1}W_{L-1}^\top + \cdots + \frac{1}{\gamma_1}W_{L-1:2}W_{L-1:2}^\top \right)W_L^\top y \right). \tag{S.88}$$

Set an $M_L \times M_L$ matrix

$$C_\gamma(W) := \sum_{i=2}^{L} \frac{\gamma_L}{\gamma_{i-1}}W_{L:i}W_{L:i}^\top. \tag{S.89}$$

Then,

$$e_L^* = y - W_L v_{L-1}^* \tag{S.90}$$

$$= y - (I + C_\gamma(W))^{-1}(W_{L:1}x + (I + C_\gamma(W))y)$$

$$= (I + C_\gamma(W))^{-1}(y - f). \tag{S.91}$$

Thus, at the fixed point, the local loss is explicitly obtained as

$$e_l^* = \frac{\gamma_L}{\gamma_l}W_{L:l+1}^\top(I + C_\gamma(W))^{-1}(y - f). \tag{S.92}$$

We can also obtain $v_l^*$. From $e_1^*$, we have

$$v_1^* = W_1 x + \frac{\gamma_L}{\gamma_1}W_{L:2}^\top(I + C_\gamma(W))^{-1}(y - f). \tag{S.93}$$

By induction, we have

$$v_l^* = e_l^* + W_l v_{l-1}^* \tag{S.94}$$

$$= W_{l:1}x + \left( \frac{\gamma_L}{\gamma_l}W_{L:l+1}^\top + \sum_{i=2}^{l-1}\frac{\gamma_L}{\gamma_{i-1}}W_{l:i}W_{L:i}^\top \right)(I + C_\gamma(W))^{-1}(y - f). \tag{S.95}$$

$\square$

### B.2.2 PROOF FOR COROLLARY 4.3

*Proof.* We consider the order of $W_{L:i}W_{L:i}^\top$ in $C_\gamma(W) = \sum_{i=2}^{L}\frac{\gamma_L}{\gamma_{i-1}}W_{L:i}W_{L:i}^\top$. Note that computing the vector $v = W_{L:i}^\top$ is equivalent to signal propagation in a deep linear network with random weights $W_{l:L-1}^T$ and an input vector $W_L^\top$. Therefore, as in Eq. (S.39), we can evaluate $\|v\|_2^2$ using kernel computations in existing studies of random neural networks. Specifically, this is equivalent to computing $h_l^\top h_l$ in the following random neural network:

$$u_k = W_k h_{k-1}, \quad h_k = \phi(u_k) \quad (k = 1, \ldots, l), \tag{S.96}$$

$$x_i \sim \mathcal{N}(0, 1/M^{a_L+b_L}) \quad (i = 1, ..., M), \tag{S.97}$$

$$W_{k,ij} \sim \mathcal{N}(0, 1/\sqrt{M}) \quad (i, j = 1, ..., M). \tag{S.98}$$

From Theorem 7.2 in Yang (2020), for any $x$, $q_k = h_k^\top h_k/M$ converges almost surely to

$$q_k = \frac{1}{\sqrt{2\pi}}\int du e^{-\frac{u^2}{2}}\phi^2(\sqrt{q_{k-1}}u), \tag{S.99}$$

in the infinite-width limit. Since we are considering a linear activation function, we have $q_l = q_{l-1}$ and

$$q_l = \frac{1}{M}x^\top x. \tag{S.100}$$

Since $x$ is an i.i.d. Gaussian vector, the law of large numbers leads to $q_l$ of $\Theta(1/M^{2(a_L+b_L)})$. For random weights given by $\mu$P i.e., $a_L + b_L = 1$, we find $W_{L:i}W_{L:i}^\top = \Theta(1/M)$. Consequently, for $\gamma_L = \Theta(M)$, $C_\gamma(W)$ remains of order 1. In contrast, for $\gamma_L = \Theta(1)$, $C_\gamma(W)$ is $O(1/M)$, causing $I + C_\gamma(W)$ to approach $I$ in the infinite-width limit. As a result, $e_l^* = \delta_l$ and $v_l^* = u_l$, and the gradient in the infinite-width limit matches that of first-order gradient descent. □

### B.2.3 BALANCE CONDITION DETERMINING $\gamma_{l<L}$

Here, we consider the order of $e_{l,1}$ with respect to $\gamma_{l<L}$. For a linear network with one-shot SI, we obtain

$$e_{l,1} = -\prod_{i=l+1}^{L} \gamma_i(\delta_l - \gamma_l W_l \delta_{l-1}) \sim 1/M^{\min(0,\bar{\gamma}_l)+\sum_{i=l+1}^{L}\bar{\gamma}_i} \tag{S.101}$$

In contrast, recall that the order of $e_l^*$ at the fixed point is

$$e_l^* \sim 1/M^{-\bar{\gamma}_l+\min(0,\bar{\gamma}_l)}. \tag{S.102}$$

Therefore, to satisfy $e_{l,1} \sim e_l^*$, the following is necessary:

$$\sum_{i=l+1}^{L} \bar{\gamma}_i = -\bar{\gamma}_l \tag{S.103}$$

for all $l < L$. This is equivalent to $\bar{\gamma}_l = 0$ for all $l < L$. Thus, $e_{l,1} \sim e_l^*$ holds if and only if $\bar{\gamma}_{l<L} = 0$.

### B.2.4 NUDGED PREDICTIVE CODING

We can extend Theorem 4.2 to the nudged PC.

**Theorem B.3.** *Suppose an L-layered linear network and put $e_l^* = v_l^* - W_l v_{l-1}^*$, where $*$ denotes the fixed point of the inference given by Eq. (S.10). The following holds:*

$$e_l^* = v_l^* - W_l v_{l-1}^* = \frac{\gamma_L}{\gamma_l} W_{L:l+1}^\top \left(I + \frac{\gamma_L}{\beta}I + C_\gamma(W)\right)^{-1} (W_{L:1}x - y) \tag{S.104}$$

$$e_L^* = v_L^* - W_L v_{L-1}^* = \left(I + \frac{\gamma_L}{\beta}I + C_\gamma(W)\right)^{-1} (W_{L:1}x - y) \tag{S.105}$$

*where $W_{L:i} = W_L W_{L-1}...W_i$.*

*Proof.* Put

$$F(v_1,...,v_L) = \beta\|y - v_L\|^2 + \sum_{l=1}^{L} \frac{\gamma_l}{2}\|v_l - W_l v_{l-1}\|^2. \tag{S.106}$$

Taking $\frac{\partial F}{\partial v_l} = 0$, we have

$$\beta(v_L - y) + \gamma_L(v_L - W_L v_{L-1}) = 0 \tag{S.107}$$

$$-\gamma_l W_l^\top(v_l - W_l v_{l-1}) + \gamma_{l-1}(v_{l-1} - W_{l-1}v_{l-2}) = 0 \quad (1 < l \le L) \tag{S.108}$$

$$-\gamma_2 W_2^\top(v_2 - W_2 v_1) + \gamma_1(v_1 - W_1 x) = 0 \quad (l = 1). \tag{S.109}$$

Putting $\chi = v_L - y$, the system of equations can be written in a matrix form as follows:

$$\begin{bmatrix} I & O & \cdots & \cdots & O \\ -W_L^\top & I & O & \cdots & O \\ O & -W_{L-1}^\top & I & \cdots & O \\ \vdots & \ddots & \ddots & & \vdots \\ O & \cdots & O & -W_2^\top & I \end{bmatrix} \begin{bmatrix} \gamma_L e_L^* \\ \gamma_{L-1}e_{L-1}^* \\ \vdots \\ \gamma_2 e_2^* \\ \gamma_1 e_1^* \end{bmatrix} = \begin{bmatrix} -\beta\chi^* \\ O \\ O \\ \vdots \\ O \end{bmatrix}. \tag{S.110}$$

From Lemma B.2, the above equation is transformed into

$$
\begin{bmatrix} e_L^* \\ e_{L-1^*} \\ \vdots \\ e_2^* \\ e_1^* \end{bmatrix} = -\beta \begin{bmatrix} \frac{1}{\gamma_L}\chi^* \\ \frac{1}{\gamma_{L-1}}W_L^\top \chi^* \\ \vdots \\ \frac{1}{\gamma_1}W_{2:L}^\top \chi^* \end{bmatrix}.
\tag{S.111}
$$

Take the following summation:

$$
e_L^* + W_L e_{L-1}^* + W_{L-1}e_{L-2}^* + \cdots + W_{L-1:2}e_1^* = v_L^* - W_{L:1}x
$$
$$
= -\frac{\beta}{\gamma_L}\left( I + C_\gamma(W) \right) \chi^*.
\tag{S.112}
$$

Thus, we can explicitly obtain $v_L$ as

$$
v_L^* = \left( \frac{\beta}{\gamma_L}I + \frac{\beta}{\gamma_L}C_\gamma(W) \right)^{-1} \left( W_{L:1}x + \frac{\beta}{\gamma_L}\left( I + C_\gamma(W) \right)y \right).
\tag{S.113}
$$

Thus, $\chi$ can be written as follows:

$$
\chi^* = y - v_L^*
\tag{S.114}
$$
$$
= y - \left( \frac{\beta}{\gamma_L}I + \frac{\beta}{\gamma_L}C_\gamma(W) \right)^{-1} \left( W_{L:1}x + \frac{\beta}{\gamma_L}\left( I + C_\gamma(W) \right)y \right)
\tag{S.115}
$$
$$
= \left( I + \frac{\beta}{\gamma_L}I + \frac{\beta}{\gamma_L}C_\gamma(W) \right)^{-1} (y - f).
\tag{S.116}
$$

From the above, we conclude that

$$
e_l^* = \frac{\beta}{\gamma_l}W_{L:l+1}^\top \left( I + \frac{\beta}{\gamma_L}I + \frac{\beta}{\gamma_L}C_\gamma(W) \right)^{-1} (y - f)
\tag{S.117}
$$
$$
= \frac{\gamma_L}{\gamma_l}W_{L:l+1}^\top \left( I + \frac{\gamma_L}{\beta}I + C_\gamma(W) \right)^{-1} (y - f)
\tag{S.118}
$$

at the fixed point.

$\square$

## C  $\mu$P OF TARGET PROPAGATION

### C.1  DERIVATION OF THEOREM 5.1

Assume that the feedback network is linear: $g_l(x) = Q_l x$. Here, we consider a reconstruction loss with L2 regularization:

$$
\mathcal{L}(Q_{l,s}) = \|Q_{l,s}\phi(W_l h_{l-1}) - h_{l-1}\|^2 + \mu_l\|Q_{l,s}\|^2
\tag{S.119}
$$

with $\mu_l \geq 0$. Note that while some work adds noise to $h_{l-1}$, it does not affect the order; therefore, we will ignore it in this derivation. As described below, by taking the ridge-less limit of $\mu$, we can evaluate the parameterization of the original TP and Difference Target Propagation (DTP) in a clear and unified manner. Considering the fixed point for $Q_L$, since $\frac{\partial l(Q_l)}{\partial Q_l} = 0$ holds, we have

$$
Q_l^* = h_{l-1}(h_l^\top h_l + \mu_l I)^{-1}h_l^\top
\tag{S.120}
$$

where $h_l = \phi(W_l h_{l-1})$. The feedback network is given by the network with Eq. (S.120). As a side note, this weight is essentially the same as the pseudo-inverse weight, which is known as an extension of the Hebbian weight (Kanter & Sompolinsky, 1987).

**Local targets of DTP.** DTP is an improved method of TP, where $\hat{h}_L$ is propagated as follows:

$$
\hat{h}_i = g_{l+1}^{\text{diff}}\left( \hat{h}_{l+1}, h_{l+1}, h_l \right) = g_{l+1}\left( \hat{h}_{l+1} \right) - \left( g_{l+1}\left( h_{l+1} \right) - h_l \right).
\tag{S.121}
$$

For the last layer, the error is given by

$$\hat{h}_L = h_L + \beta(y - h_L) \tag{S.122}$$

For a linear feedback network, we have

$$\hat{h}_l = g_{l+1}\left(\hat{h}_{l+1}\right) - (g_{l+1}(h_{l+1}) - h_l) \tag{S.123}$$

$$= h_l + Q_{l+1}(\hat{h}_{l+1} - h_{l+1}) \tag{S.124}$$

$$= h_l + Q_{l+1}((h_{l+1} + Q_{l+2}(\hat{h}_{l+2} - h_{l+2})) - h_{l+1}) \tag{S.125}$$

$$= h_l + Q_{l+1}Q_{l+2}(\hat{h}_{l+2} - h_{l+2}) \tag{S.126}$$

$$= h_l - \beta \prod_{i=l+1}^{L} Q_i \delta_L. \tag{S.127}$$

Therefore, at the equilibrium point for $Q_l$, for $l \leq L - 2$, we have

$$\hat{h}_l - h_l = -\beta \prod_{i=l+1}^{L} Q_i^* \delta_L \tag{S.128}$$

$$= -\beta h_l \prod_{i=l+1}^{L-1} (h_i^\top h_i + \mu_l I)^{-1} h_i^\top h_i (h_L^\top h_L + \mu_l I)^{-1} h_L^\top \delta_L \tag{S.129}$$

$$= -\beta h_l \prod_{i=l+1}^{L-1} (K_i^A + \mu_i' I)^{-1} K_i^A (K_L^A + \mu_L' I)^{-1} h_L^\top \delta_L \tag{S.130}$$

where $\delta_L = \partial\mathcal{L}/\partial f$ and for $l = L - 1$, we have $\hat{h}_{L-1} - h_{L-1} = -\beta h_{L-1}(K_L^A + \mu_L' I)^{-1} h_L^\top \delta_L$. To avoid an uninteresting change of order, we introduce $\mu_l = \mu_l'/M^{\bar{\mu}_l}$ and require that it have the same order as $K_i^A$. This is essentially equivalent to the valid condition argued in Ishikawa & Karakida (2024), which requires the damping term to have the same order as the preconditioner in the second-order optimization. We note that we can take the ridge-less limit $\mu_l' \to 0+$ because $K_i^A$ (S.39) is typically set to be regular at random initialization in the neural tangent kernel literature (Jacot et al., 2018; Yang, 2020). For instance, this holds true for normalized input samples with $\|x\| = 1$.

**Local targets of original TP.** The signal propagation in the feedback network is

$$\hat{h}_l = Q_{l+1}^* \cdots Q_L^* \hat{h}_L \tag{S.131}$$

$$= h_l \prod_{i=l+1}^{L-1} (h_i^\top h_i + \mu_i I)^{-1} h_i^\top h_i (h_L^\top h_L + \mu_L I)^{-1} h_L^\top (h_L - \beta \delta_L) \tag{S.132}$$

$$\to h_l - \beta h_l \prod_{i=l+1}^{L-1} (K_i^A)^{-1} K_i^A (K_L^A)^{-1} h_L^\top \delta_L \quad (\mu_l' \to 0+). \tag{S.133}$$

Thus, the target is reduced to essentially the same as that in DTP (S.130) and we can treat both in the same manner.

**On Condition A.1.** The update for the last layer is identical to that of SGD with BP, thus

$$\partial_{\eta'} \Delta W_L h_{L-1,1}\big|_{\eta'=0} = -\frac{1}{M^{\theta_L-1}} \beta \delta_L K_{L-1}^A \tag{S.134}$$

Next, we consider the $L - 1$ layer.

$$\partial_{\eta'} \Delta W_{L-1} h_{L-2,1}\big|_{\eta'=0} = \frac{1}{M^{\theta_{L-1}}} (\hat{h}_{L-1} - h_{L-1}) h_{L-1}^\top h_{L-1} \tag{S.135}$$

$$= -\frac{1}{M^{\theta_{L-1}-1}} \beta h_{L-1}(h_L^\top h_L + \mu_L I)^{-1} h_L^\top \delta_L K_{L-1}^A \tag{S.136}$$

$$= -\frac{1}{M^{\theta_{L-1}-1}} \beta h_{L-1}(K_L^A + \mu_L' I)^{-1}(K_L^A - M^{-1} h_L^\top y) K_{L-1}^A. \tag{S.137}$$

Similarly, when $\hat{h}_l - h_l = -\beta \prod_{i=l+1}^{L} Q_i \delta_L$, we have

$$\partial_{\eta'} \Delta W_l h_{l-1,1}\big|_{\eta'=0} = \frac{1}{M^{\theta_l}} (\hat{h}_l - h_l) h_{l-1}^\top h_{l-1} \tag{S.138}$$

$$= -\frac{1}{M^{\theta_l-1}} \beta h_l \prod_{i=l+1}^{L-1} (h_i^\top h_i + \mu_i I)^{-1} h_i^\top h_i (h_L^\top h_L + \mu_L I)^{-1} h_L^\top \delta_L K_{l-1}^A \tag{S.139}$$

$$= -\frac{1}{M^{\theta_l-1}} \beta h_l \prod_{i=l+1}^{L-1} (K_i^A + \mu_i' I)^{-1} K_i^A (K_L^A + \mu_L' I)^{-1} (K_L^A - M^{-1} h_L^\top y) K_{l-1}^A \tag{S.140}$$

for $l = 1, ..., L-2$. On the right-hand side, $h_L \sim 1/M^{a_L+b_L-1/2}$, and from Eq. (S.39), we have

$$K_L^A - M^{-1} h_L^\top y \sim \max\{1/M^{2(a_L+b_L)}, 1/M^{a_L+b_L+1/2}\} \tag{S.141}$$

$$= 1/M^{a_L+b_L+1/2}. \tag{S.142}$$

In the last line, we used $a_L + b_L \geq 1/2$. Here, from Assumption A.1, which states that $a_L + b_L \geq 1/2$, we obtain

$$r_l = \begin{cases} \theta_1 - a_L - b_L + 1/2 & (l=1), \\ \theta_l - 1 - a_L - b_L + 1/2 & (1 < l < L), \\ \theta_L - 1 & (l = L). \end{cases} \tag{S.143}$$

**On Condition A.2.**

$$\partial_{\eta'} (W_{L,0} \Delta h_{L-1,1})\big|_{\eta'=0} = \frac{1}{M^{\theta_{L-1}}} W_{L,0} \mathrm{diag}(\phi'_{L-1})(\hat{h}_{L-1} - h_{L-1}) h_{L-2}^\top \tag{S.144}$$

$$= -\frac{1}{M^{\theta_{L-1}-1}} \beta h_L (h_L^\top h_L + \mu_L' I)^{-1} h_L^\top \delta_L K_{L-2}^A \tag{S.145}$$

$$= -\frac{1}{M^{\theta_{L-1}-1}} \beta h_L (K_L^A + \mu_L' I)^{-1} (K_L^A - M^{-1} h_L^\top y) K_{L-2}^A. \tag{S.146}$$

Thus, its order is

$$\partial_{\eta'} (W_{L,0} \Delta h_{L-1,1})\big|_{\eta'=0} \sim 1/M^{\theta_{L-1}-1+(a_L+b_L-1/2)-2(a_L+b_L)+(a_L+b_L+1/2)} \tag{S.147}$$

$$= 1/M^{\theta_{L-1}-1=r_{L-1}+(a_L+b_L)-1/2}. \tag{S.148}$$

Finally, from Conditions A.1 and A.2, the $\mu$P is given by

$$\theta_1 - a_L - b_L + 1/2 = 0 \quad (l=1), \tag{S.149}$$
$$\theta_l - a_L - b_L - 1/2 = 0 \quad (1 < l < L), \tag{S.150}$$
$$\theta_L - 1 = 0 \quad (l = L), \tag{S.151}$$
$$a_L + b_L - 1/2 = 0. \tag{S.152}$$

$\square$

### C.1.1 Disappearance of Kernel regime

**Corollary C.1.** *Stable learning satisfying Condition A.2 leads to $r_{L-1} = 0$ for TP and DTP.*

*Proof.* From Eq. (S.148), we have

$$r_{L-1} + (a_L + b_L) - 1/2 = 0. \tag{S.153}$$

From Eq. (S.14), $a_L + b_L \geq 1/2$ and we have $r_{L-1} \geq 0$. In contrast, from the stability of learning, we have $r_{L-1} \leq 0$. Thus, $r_{L-1} = 0$. $\square$

Note that, precisely speaking, $r_{L-1} = 0$ does not necessarily imply $r_{l<L-1} = 0$. However, it is often considered unnatural (or uninteresting) to examine cases in which the progress of learning depends on individual layers. Therefore, the $\mu$P typically assumes a uniform parameterization, meaning $r_{l<L} = r$ (see Theorem G.4 of Yang & Hu (2021)). In this sense, $r_{L-1} = 0$ indicates the disappearance of the kernel regime.

One might be surprised by the fact that $b_L = 1/2$ is allowed in the feature learning and that the kernel regime disappears. Note that the feedback weight in the last layer (S.120) essentially differs from $W_L$. The feedback weight receives $h_L$ as input whereas $W_l$ receives $h_{L-1}$. This makes

$$Q_L \sim 1/M^{1/2-(a_L+b_L)} \tag{S.154}$$

$W_L \sim 1/M^{a_L+b_L}$. The gradient is proportional to $Q_L$ in TP and $W_L$ in BP. The feedback weight contributes more significantly to TP's gradient when $a_L + b_L \geq 1/2$. This eventually makes the index $r$ of the hidden layer (S.143) get quite large even for $a_L + b_L = 1/2$. We also need to be careful about the order of condition A.2 (S.148). Because the feedback weight (S.120) has a lower alignment exponent (Everett et al., 2024), this causes the condition 2 of TP to be smaller than that of SGD (or K-FAC), i.e., $1/M^{r_{L-1}+(a_L+b_L)-1}$. Therefore, stable feature learning is possible even for $a_L + b_L = 1/2$.

**Remark on zero head initialization.**   Related to the size of $b_L$, the parameter initialization with $b_L > 1/2$ ($b_L > 1$ for SGD) reduces to the $\mu$P of $b_L = 1/2$ ($b_L = 1$ for SGD) because $W_{l,0}$ becomes negligible compared to $\Delta W_{l,0}$. To illustrate the intuition, let us introduce the case where the weight of the last layer in a feedforward network is initialized as $W_L = O$, that is, $b_L = \infty$.

In this case, only the last layer is updated during the first step because $Q_L^* = O$ does not propagate the local error to the downstream layers. After the first-step update, the weight is given by

$$W_{L,1} = -\frac{\eta'}{M^{\theta_L}} \delta_L h_{L-1}^\top. \tag{S.155}$$

and $W_{l<L,1} = W_{l<L,0}$. Thus,

$$\partial_{\eta'} \Delta W_l h_{l-1,1}\big|_{\eta'=0} = -\frac{1}{M^{\theta_L}} \delta_L h_{L-1}^\top h_{L-1} \tag{S.156}$$

$$= -\frac{1}{M^{\theta_L-1}} \delta_L K_{L-1}^A. \tag{S.157}$$

and

$$h_{L,1} = \Theta(1/M^{\theta_L-1}), \quad K_L^A = \Theta(1/M^{2\theta_L-1}), \tag{S.158}$$

Substituting these into Eqs. (S.137,S.140), we obtain

$$r_l = \begin{cases} \theta_1 - \theta_L + 1 & (l = 1), \\ \theta_l - \theta_L & (1 < l < L), \\ \theta_L - 1 & (l = L). \end{cases} \tag{S.159}$$

From Eq. (S.146),

$$\partial_{\eta'}(W_{L,0} \Delta h_{L-1,1})\big|_{\eta'=0} = \Theta(1/M^{\theta_{L-1}-1}). \tag{S.160}$$

Finally, Conditions A.1 and A.2 lead to

$$\theta_1 - (\theta_L - 1) = 0 \quad (l = 1), \tag{S.161}$$
$$\theta_l - 1 - (\theta_L - 1) = 0 \quad (1 < l < L), \tag{S.162}$$
$$\theta_L - 1 = 0 \tag{S.163}$$

Thus, the $\mu$P is the same as in the case of random head initialization.

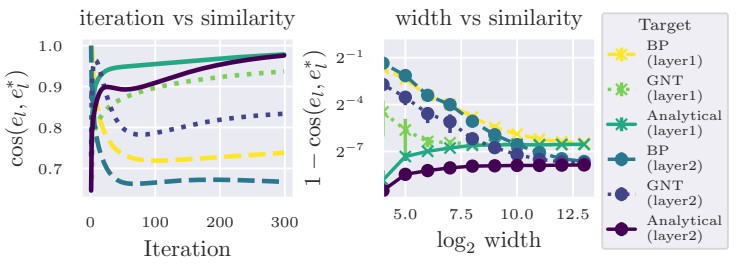

**Figure S.1 : During the training of the linear network, it converges to the analytical solution.** We trained a 3-layer linear network using synthetic data.

## C.2 STABLE PARAMETERIZATION FOR FEEDBACK NETWORK

The feedback network minimizes the following loss function:

$$L(Q_l) = \frac{1}{2M_{l-1}} \|\phi(Q_l h_l) - h_{l-1}\|^2. \tag{S.164}$$

where dividing by $M_{L-1}$ is to ensure that $L(Q_l) = \Theta(1)$, which is the default setting in PyTorch. We consider the parameterization in the feedback network:

$$Q_l \sim \mathcal{N}(0, \sigma'^2/M^{2\bar{q}_l}), \quad \tau_l = \frac{\tau_l'}{M^{\bar{\tau}_l}}, \tag{S.165}$$

where $\tau_l$ denotes the learning rate for the feedback network.

To ensure that the update $\Delta Q_l h_l = \Theta(1)$ holds, we have

$$\Delta Q_l h_l = \frac{\tau'}{M^{\bar{\tau}_l+1}} \left(\phi(Q_l h_l) - h_{l-1}\right) \phi'(Q_l h_l) h_l^\top h_l. \tag{S.166}$$

Here, because $h_{l<L}^\top h_{l<L} = \Theta(M)$ and $h_L^\top h_L = \Theta(1/M^{2b_L-1})$, assuming $\Delta Q_l h_l = \Theta(1/M^{r_l})$, we obtain:

$$r_l = \begin{cases} \bar{\tau}_l & (1 < l < L), \\ \bar{\tau}_L + 2b_L & (l = L) \end{cases} \tag{S.167}$$

Therefore,

$$\bar{\tau}_{l<L} = 0, \quad \bar{\tau}_L = -2b_L. \tag{S.168}$$

If we optimize the feedback network for one step, we have

$$Q_{l<L,1} = Q_{l<L,0} - \frac{\tau'}{M^{\bar{\tau}_l+1}} h_{l-1} h_l^\top \sim \frac{1}{M^{\min(1,q_l)}}, \tag{S.169}$$

$$Q_{L,1} = Q_{L,0} - \frac{\tau'}{M^{\bar{\tau}_L+1}} h_{L-1} h_L^\top \sim \frac{1}{M^{\min(1/2-b_L,q_L)}}. \tag{S.170}$$

And,

$$Q_{l<L}^* = h_{l-1}(h_l^\top h_l + \mu I)^{-1} h_l^\top \sim \frac{1}{M}, \tag{S.171}$$

$$Q_L^* = h_{L-1}(h_L^\top h_L + \mu I)^{-1} h_L^\top \sim \frac{1}{M^{1/2-b_L}}. \tag{S.172}$$

Therefore, in this case, $Q_{l,1} = Q_l^*$ holds when

$$\bar{q}_{l<L} \geq 1, \quad \bar{q}_L \geq 1/2 - b_L. \tag{S.173}$$

## D ADDITIONAL EXPERIMENTS

### D.1 PREDICTIVE CODING

#### D.1.1 LINEAR NETWORK

In Figure S.1 , we measure the similarity of the inference vector with BP, GNT, and the analytical solution. With fewer inference iterations, the model behaves more like BP; however, as the number of iterations increases, the model converges toward the analytical solution. Furthermore, as the middle layer width $M_l$ increases, the gap between GNT and BP decreases. Figure S.2 further demonstrates that reducing the output dimension $M_L$ brings the model closer to BP. However, increasing $M_L$ moves the model away from BP, though this divergence is more gradual as $\bar{\gamma}_L$ approaches zero.

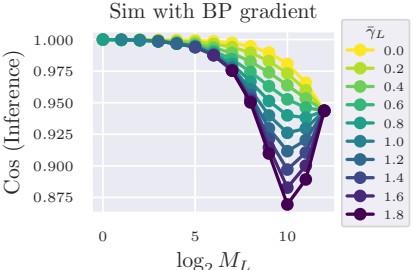

**Figure S.2 : As $M_L$ approaches 1, the update vector in PC converges to that of BP.** We conducted inference training on a 3-layer linear network and measured the similarity between PC and BP. The results demonstrate that PC approaches BP as $M_L$ decreases and $\bar{\gamma}_L$ increases.

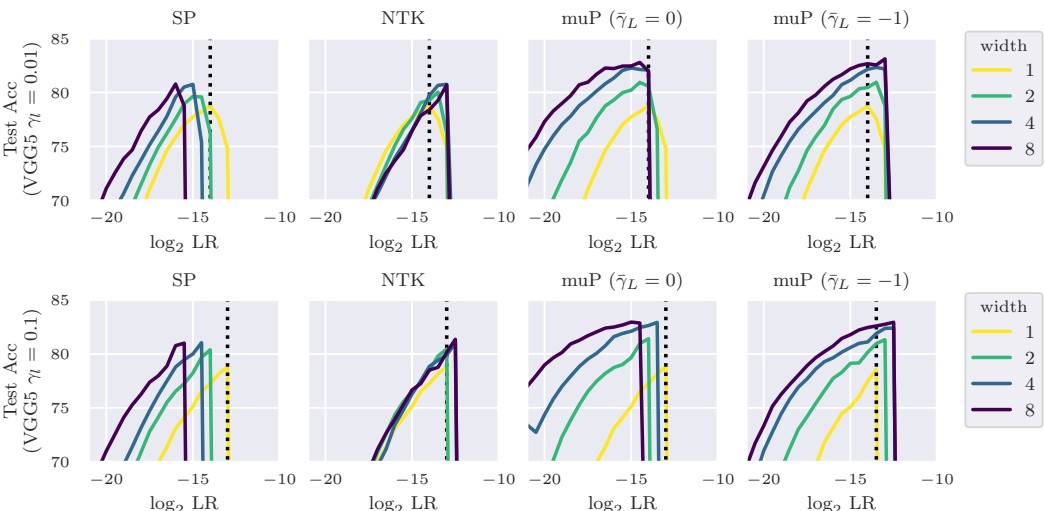

**Figure S.3 : In VGG5, the learning rate also transfers across widths.** In SP, the optimal learning rate shifts based on model width, whereas in $\mu$P, it remains fixed. Additionally, we trained with two different $\gamma_l$ values, and under $\mu$P ($\bar{\gamma}_L = -1$), the learning rate consistently transfers across widths, regardless of $\gamma_l$. The model was trained for 40 epochs on 1024 samples from FashionMNIST.

### D.1.2   ADDITIONAL EXPERIMENTS ON $\mu$ TRANSFER FOR PC

**Architecture**   In the main text, we primarily focused on MLP and CNN. However, our $\mu$P is architecture-independent. The results for VGG5 are presented in Figure S.3 . Furthermore, the $\mu$Transfer observed in Figure 4 also holds for MLP, as demonstrated in Figure S.4 .

**Loss Type**   In the main text, we mainly used mean squared error (MSE) loss. However, this can be replaced with cross-entropy (CE) loss. As demonstrated in Figure S.5 , $\mu$P for PC also transfers the learning rate across widths when using cross-entropy loss.

**Optimizer**   In this paper, we primarily focus on weight updates using SGD. However, it is also possible to update the weights using Adam instead of SGD. In this case, the corresponding $\mu$P is as follows:

$$\begin{cases} b_1 = 0, & b_{1<l<L} = 1/2, & b_L = 1, \\ c_1 = 0, & c_{l>1} = 1. \end{cases} \tag{S.174}$$

For Adam, the scaling of $b_l$ and $c_l$ does not depend on $\bar{\gamma}_L$. Additionally, in Adam, the gradients are normalized, which means that $\mu$P remains unchanged regardless of whether the gradients are generated by BP or PC. When considering the stability of the inference, scaling with respect to $\bar{\gamma}_L$ can be treated in the same manner as in the case of SGD.

$$\bar{\gamma}_{l<L} = 0, \quad \bar{\gamma}_L = 1. \tag{S.175}$$

**Train sample**   We reduced the number of training samples in most of the graphs for $\mu$Transfer. By reducing the number of training samples, finite-width models are known to behave more similarly to

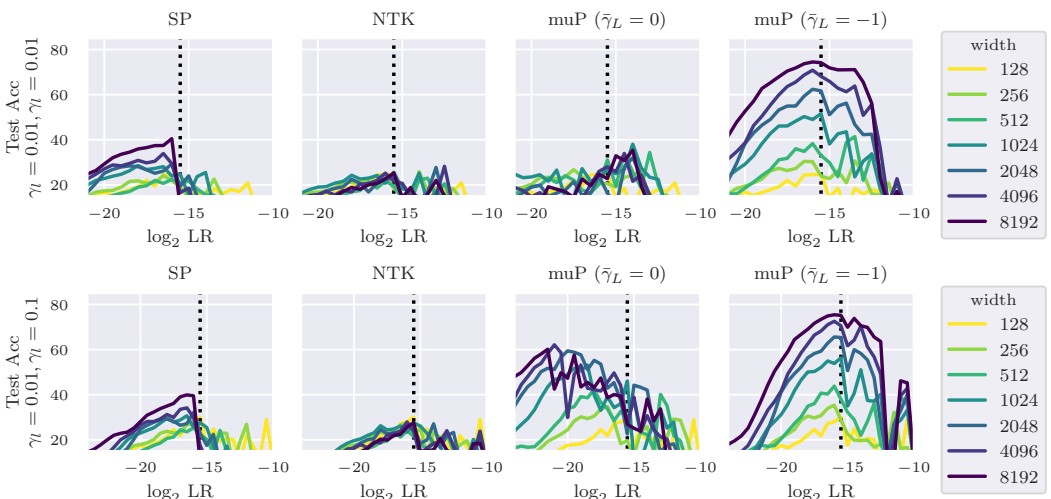

**Figure S.4 : Without F-ini, $\mu$P with $\bar{\gamma}_L = -1$ transfers the learning rates across widths also in MLP.** We trained a 3-layer MLP on FashionMNIST without F-ini.

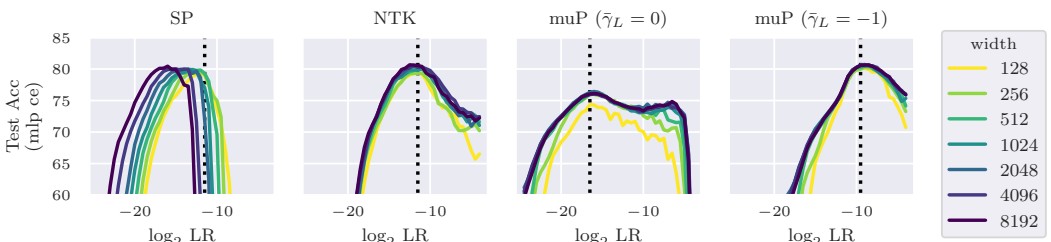

**Figure S.5 : $\mu$P for PC transfers learning rates across widths even with Cross Entropy.** We train 3-layer MLP for 40 epochs on 1024 samples from FashionMNIST.

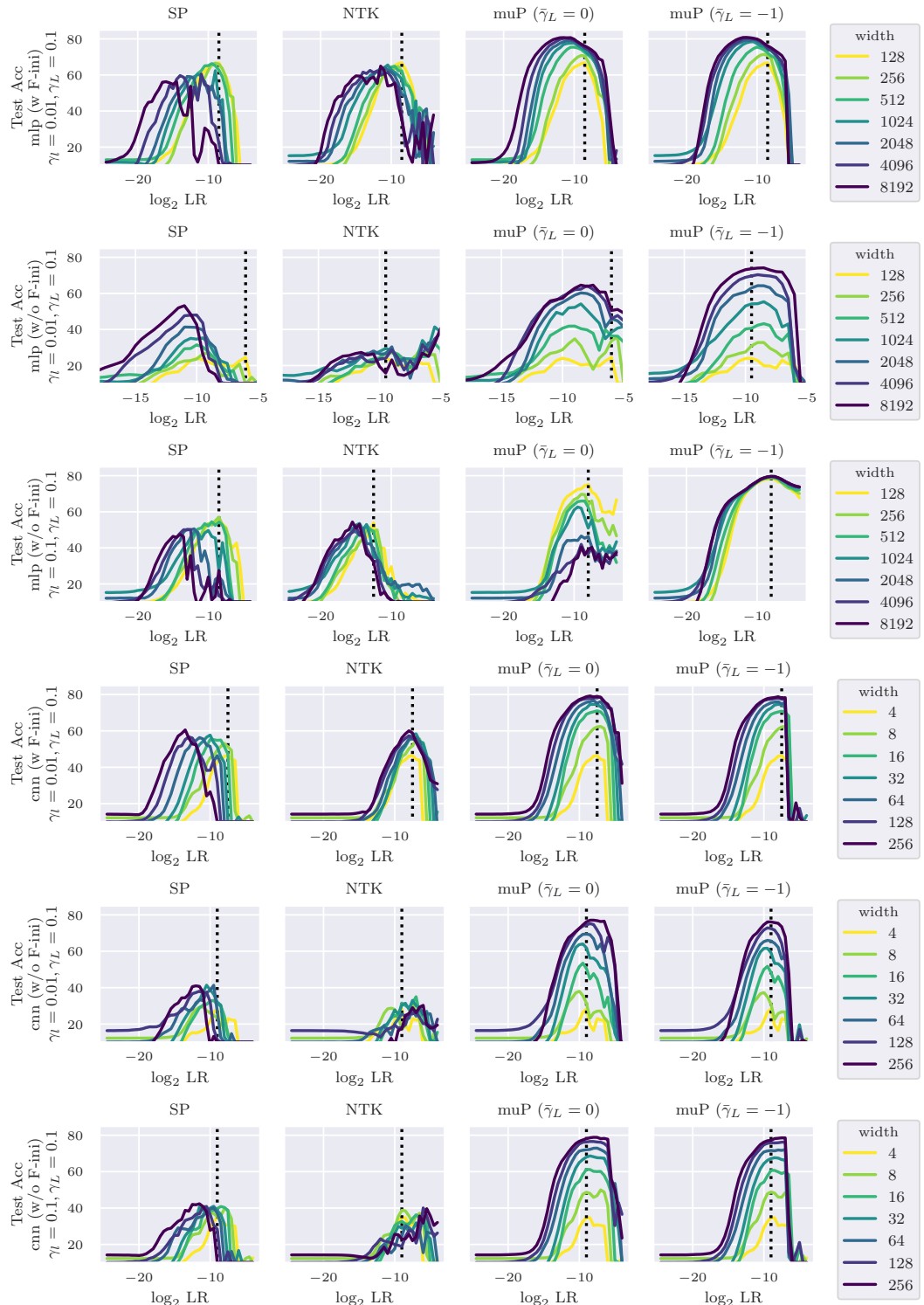

**Figure S.6 :** $\mu$**P with** $\bar{\gamma}_L = -1$ **can constantly transfer the learning rates across width** We confirmed $\mu$Transfer when training PC with Adam to update parameters in both MLP and CNN. In the training of MLP without F-ini, we observe that $\mu$P with $\bar{\gamma}_L = -1$ consistently stabilizes training and performs well. All experiments were conducted on FashionMNIST with 1024 samples.

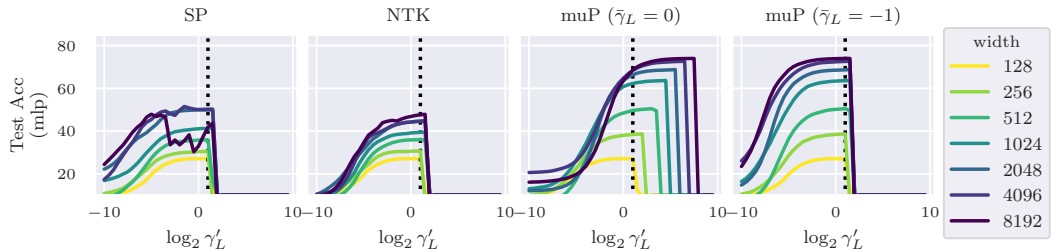

**Figure S.7 : When training with Adam, muP with $\bar{\gamma}_L = -1$ transfer $\gamma_L$ across width.** We trained a 3-layer MLP on FashionMNIST with Adam.

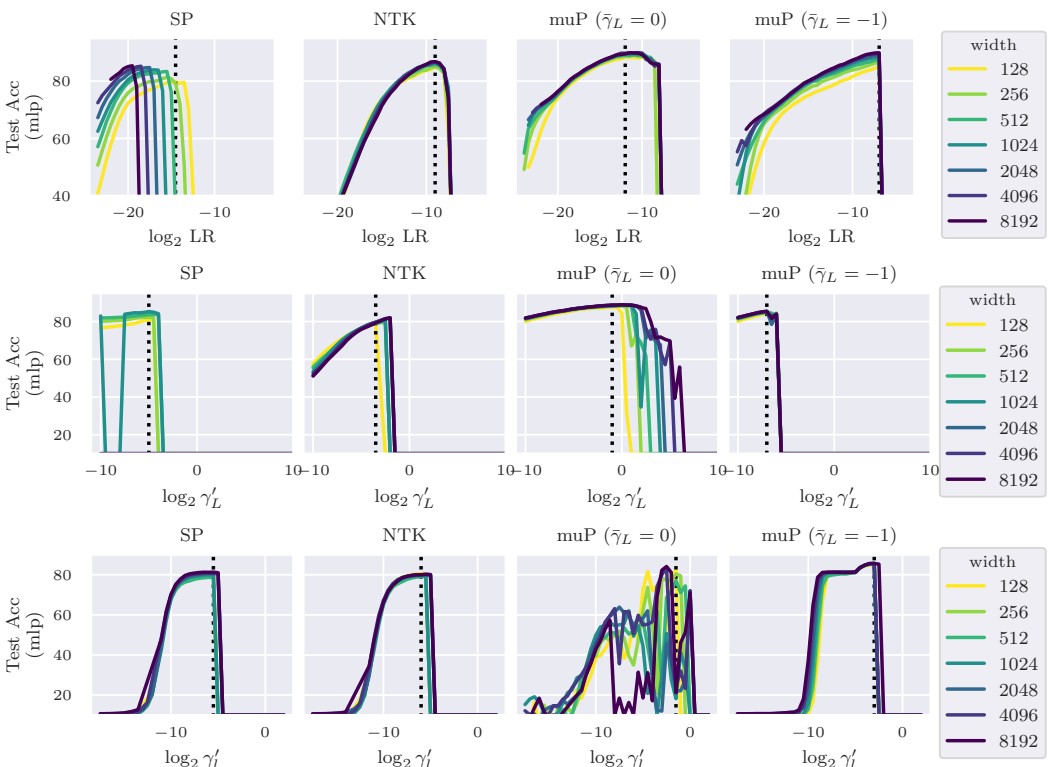

**Figure S.8 : The results of $\mu$P for PC are independent of the number of training samples.** We train a 3-layer MLP on FashionMNIST with full training samples. The stability of $\mu$P holds even with all training samples.

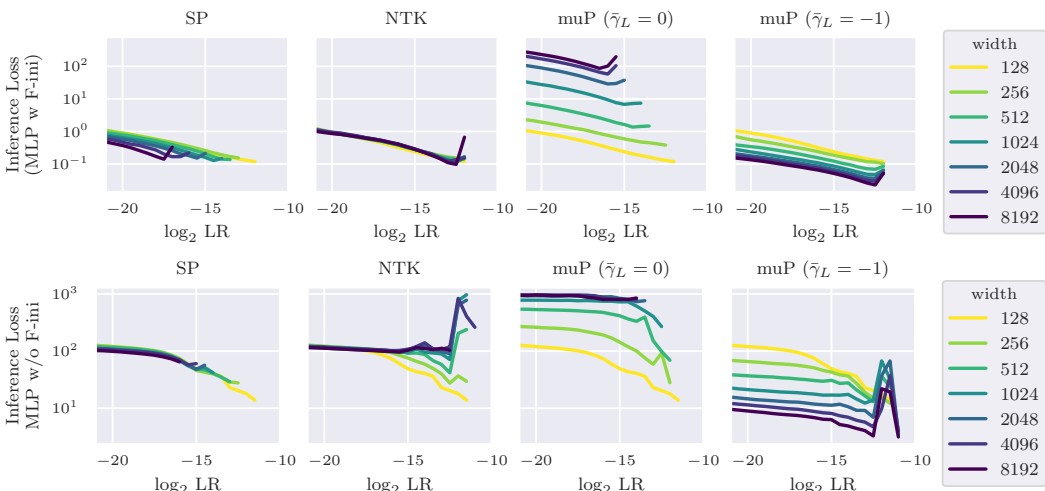

**Figure S.9 : When evaluating the loss after inference, only $\mu$P with $\bar{\gamma}_L = -1$ satisfies the empirical rule of "wider is better"** Regardless of whether F-ini is applied, $\mu$P with $\bar{\gamma}_L = -1$ consistently reduces the loss during inference with stability. We trained a 3-layer MLP on FashionMNIST.

infinite-width models, as has often been seen in papers examining the theoretical aspects of feature learning (Geiger et al., 2020; Ishikawa & Karakida, 2024). However, even when training on the full dataset, $\mu$P remains stable across widths, as shown in Figure S.8 .

**Inference Loss** When considering the stability of inference, we can observe the loss before updating the parameters after inference. As shown in Figure S.9 , when training a 3-layer MLP on FashionMNIST, only $\mu$P with $\bar{\gamma}_L = -1$ consistently reduces the inference loss as the model width increases."

**Base width and inference iterations** In $\mu$-transfer, some research set the width of the smaller model used for tuning the learning rate, as the base width, denoted by $M'$, and adjusts the learning rate using $\eta_l = \eta'_l/(M/M')^{c_l}$. As shown in Figure S.10 , the choice of $M'$ (a smaller $M'$) can sometimes make $\mu$P with $\bar{\gamma}_L = 0$ more sensitive.

As shown in Figure S.11 , the shift in the optimal learning rate at $\gamma_L = 0$ with $M' = 128$ becomes more evident as the number of inference iterations increases. This is likely because, with more iterations, the dynamics of inference play a more critical role in weight updates. In summary, to achieve stable $\mu$Transfer independent of the base width and the number of inference iterations, we should use $\mu$P with $\gamma_L = -1$.

**Sequential Inference and Synchronous Inference** In the main text, we focused on Sequential Inference, where $u_l$ is updated layer by layer, starting from the output layer. However, Synchronous Inference, where all layers are updated simultaneously, can also be considered. For the differences between Sequential Inference and Synchronous Inference, see Algorithm.1. As shown in Figure S.12 , since $\mu$P for PC is validated at fixed points, it is also applicable to Synchronous Inference.

**Additional Experiments with Figure 3** Figure S.13 presents the results of the same experiment shown in Figure 3 (Right), but with the CIFAR-10/CIFAR=100 dataset and the VGG5 model. It is evident that even with CIFAR-10, CIFAR-100 and VGG5, $\mu$P achieves higher accuracy compared to SP and NTK.

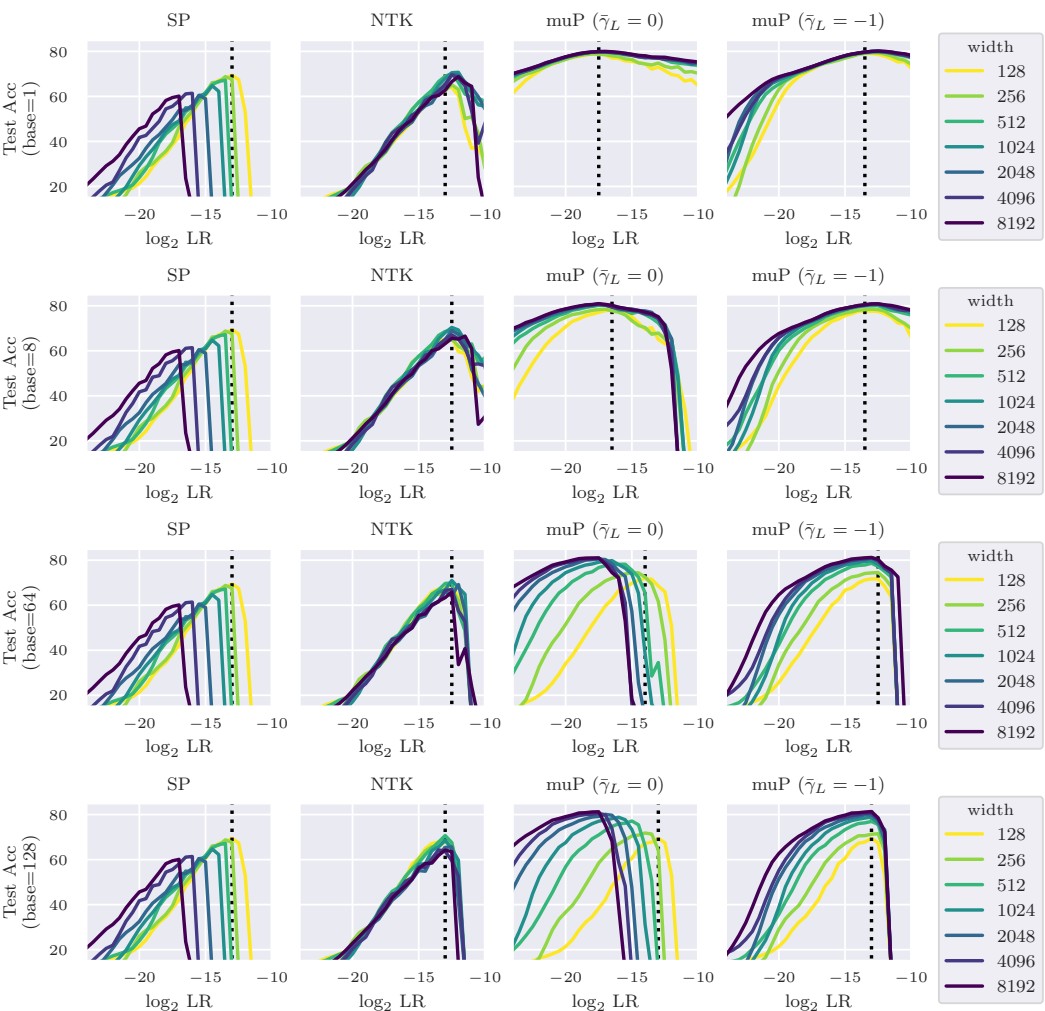

**Figure S.10 : When the base width $M'$ is large, $\mu$P with $\bar{\gamma}_L = 0$ tends to fail with $\mu$transfer.** We train a 3-layer MLP on FashionMNIST. This suggests that $\mu$P with $\gamma_L = -1$ should be used when setting the base width, even with F-ini. The inference is performed for 100 iterations.

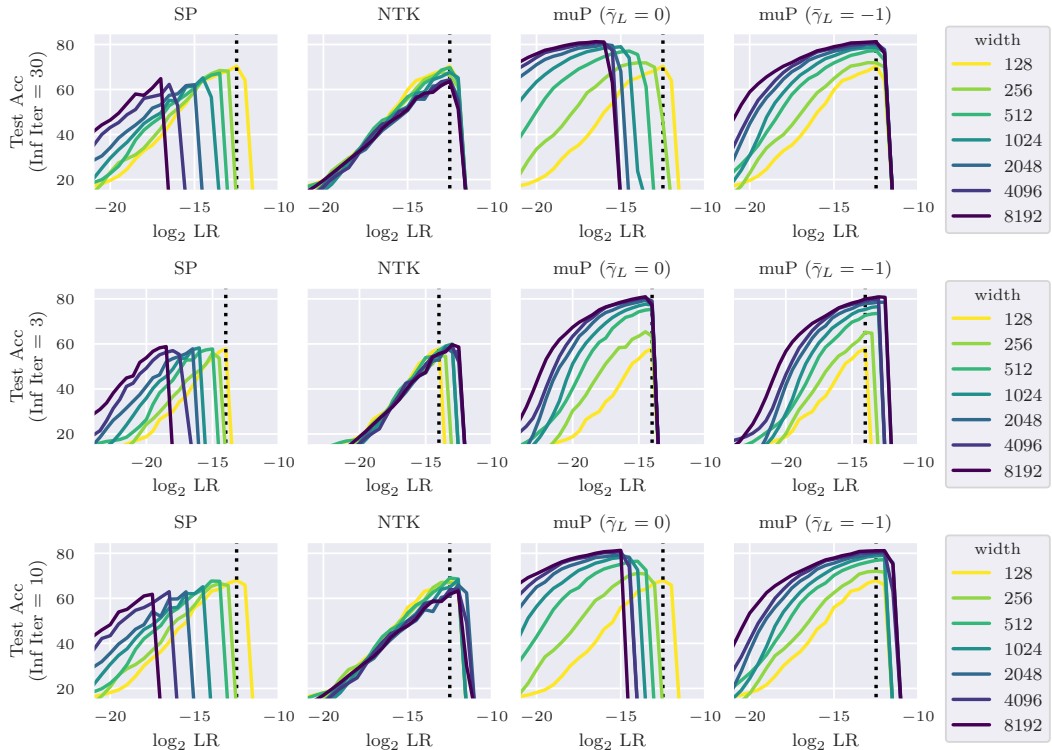

**Figure S.11 :** $\mu$P with $\bar{\gamma}_L = -1$ **maintains high inference stability and successfully performs** $\mu$**-transfer even with a large number of inference iterations.** We conducted the experiment shown in Figure S.10 with varying numbers of inference iterations. Even with a larger number of inference iterations, $\mu$P with $\gamma_L = -1$ consistently transfers the learning rate across different widths.

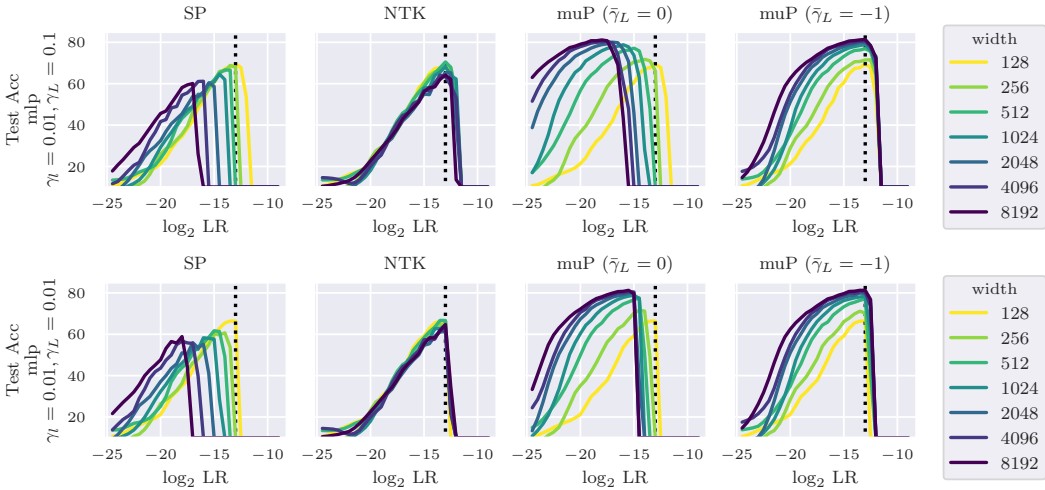

**Figure S.12 :** $\mu$P for PC also transfers learning rates across widths in synchronous inference. $\mu$P for PC can also be applied in synchronous inference. Note that when the base width is set to 128, as in SI, the learning rate does not transfer in $\mu$P with $\bar{\gamma}_L = 0$.

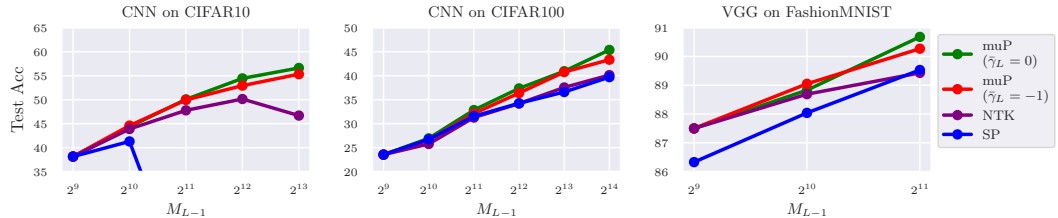

**Figure S.13 :** $\mu$**P scales better than SP and NTK.** We trained a CNN by PC with F-ini on CIFAR10/CIFAR100 and a VGG5 on the full FashionMNIST dataset. The "wider is better" principle holds for $\mu$P.

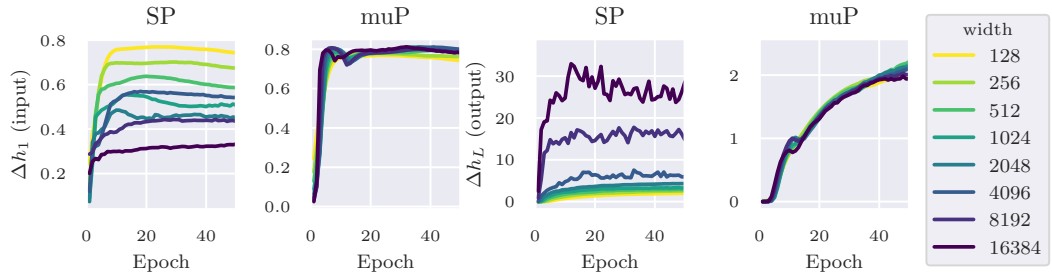

**Figure S.14 : In Target Propagation, using** $\mu$**P ensures that** $\Delta h_l$ **remains consistent across widths.** This figure shows the RMS norm of $\Delta h_l$ during training. For SP, $\Delta h_l$ in the input layer diminishes as the width increases, while $\Delta h_l$ in the output layer diverges with increasing width. Consequently, the training dynamics become unstable. In contrast, with $\mu$P, $\Delta h_l$ remains consistent across different widths in both the input and output layers.

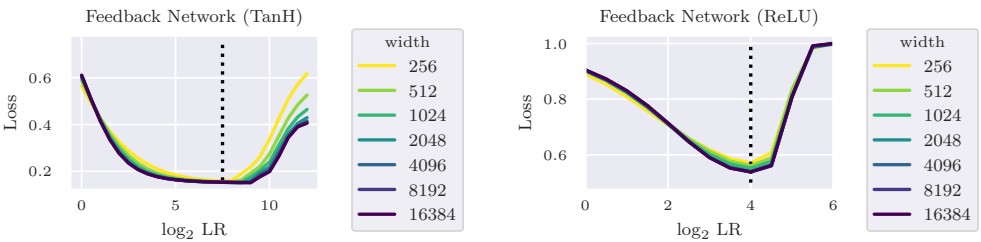

**Figure S.15 : Learning rate transfer in Feedback networks.** We demonstrate that the learning rate in feedback networks transfers effectively across widths using toy data. Both the feedforward and feedback networks include a Tanh/ReLU activation function following the linear layer.

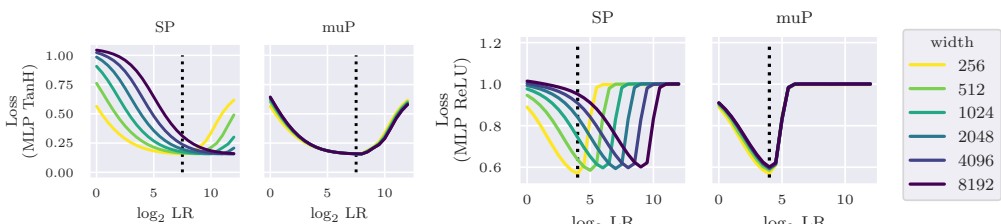

**Figure S.16 : Learning rate transfer in Feedback networks (output layer).** We show that the learning rate in feedback networks transfers across widths using toy data. Unlike the hidden layers, the learning rate in the output layer does not transfer under the default setting, which requires $\mu$P scaling. Both the feedforward and feedback networks include a Tanh/ReLU activation function after the linear layer.

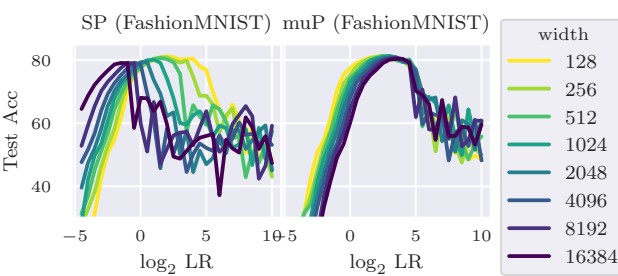

**Figure S.17 : Even when training the feedback network with DRL, $\mu$P demonstrates greater stability compared to SP.** We trained a 3-layer MLP on the FashionMNIST using DRL. While SP exhibits a shift in the maximum learning rate as the model width increases, $\mu$P consistently transfers the optimal learning rates across different widths.

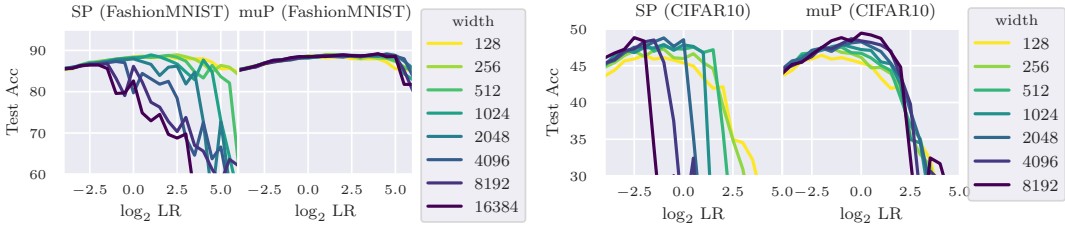

**Figure S.18 : $\mu$P for TP remains stable regardless of the dataset or the number of training samples.** We trained a 3-layer MLP on both FashionMNIST and CIFAR-10 using the full training samples. With $\mu$P, the learning rate successfully transfers across widths, ensuring that the maximum learning rate remains consistent regardless of the model width.

## D.2 TARGET PROPAGATION

### D.2.1 ADDITIONAL EXPERIMENTS ON $\mu$TRANSFER FOR TP

**Temporal change of activation** In Figure S.14 , we observed $\Delta h$ during the training of an MLP on FashionMNIST. SP exhibits a dependency of $\Delta h$ on width, whereas $\mu$P demonstrates consistent behavior, independent of width.

**Feedback Network** As discussed in Section C.2, stable parameterization is crucial not only for feedforward but also for feedback networks. We verified this with $\mu$P, as shown in Figures S.15 and S.16 .

**DRL** Meulemans et al. (2020) proposes the difference reconstruction loss (DRL) for constructing feedback networks. In Figure S.17 , we empirically confirm that our $\mu$P works effectively with DRL when training an MLP on FashionMNIST.

**Training samples** As with PC, in the case of TP, Figure 6 uses 1024 training samples. Similar results were observed when using the full training dataset, as shown in Figure S.18 .

## E  EXPERIMENTAL SETTINGS

**Architecture and dataset**    We trained the following three models:

- **MLP:** We trained a 3-layer multilayer perceptron (MLP) with Tanh activation. The MLP models do not include bias.
- **CNN:** We trained a 3-layer CNN with Tanh activation. The models consist of two-layer convolutional layers and a linear layer. We trained with different hidden widths where the width is proportional to the input dimension of the output layer. (For example, when the width is set to 4, the input dimension of the final layer is 512.) Max pooling is applied after the activation function.
- **VGG5:** We trained a VGG-like model consisting of 4 convolutional blocks and 3 linear blocks, based on the structure described in (Pinchetti et al., 2024). When the width is set to 8, it matches the VGG5 model in Pinchetti et al. (2024), with the channel sizes being [128, 256, 512, 512].

**Dataset and batch size**    We used FashionMNIST and CIFAR-10 datasets without applying any data augmentation. The settings for batch size and training samples were as follows:

- **PC** In the experiments on $\mu$Transfer, FashionMNIST was generally trained with 1024 training samples and a batch size of 1024, except for Figure S.8 . However, when training VGG5, the batch size was reduced to 64 due to memory constraints. In the experiment verifying the scaling of $\mu$P with respect to width (Figure 3), all training samples were used, with a batch size of 1024.
- **TP** In Figures S.14  and 6, we trained a 3-layer MLP using 1024 training samples. Note that in Figure S.18  in the Appendix, FashionMNIST, and CIFAR-10 were trained using the full datasets. For the activation function of feedback networks, the same activation function as one used in the forward pass is utilized (i.e., $\psi = \phi$).

**Training recipe**    Weight decay was not applied during the parameter updates for feedforward networks. For SGD, the momentum was set to 0.9, and for AdamW, the parameters $(\beta_1, \beta_2)$ were set to $(0.9, 0.99)$.

- **PC** The reduction mode for the loss function was set to "sum" to align the order of all terms in the free energy function.
- **TP** For feedback networks, weight decay was set to $10^{-4}$ and the learning rate for the target was set to $\hat{\eta} = 0.01$. Before starting the main training, only the feedback network was trained for 5 epochs with the feedforward network fixed.

