# OpenReview forum: "Local Loss Optimization in the Infinite Width: Stable Parameterization of Predictive Coding Networks and Target Propagation"
_ICLR.cc/2025/Conference — ICLR 2025 Poster_

### Official Review · Reviewer_ubuK · 2024-10-24

**Soundness:** 3
**Presentation:** 1
**Contribution:** 3
**Rating:** 6
**Confidence:** 2

**Summary:**

This paper considers maximal update parameterization for local learning, an alternative to backpropagation for optimizing neural networks. In particular, the authors consider predictive coding and target propagation.

The authors develop theory to show that hyperparameters can be transferred between networks of differing widths, matching existing guarantees for backpropagation, which helps reduce the amount of computation required for training with local learning. Further, they prove that under certain assumptions, target learning reduces to gradient descent, which implies that backpropagation and local learning will converge to the same model.

**Strengths:**

Originality: I’m not so familiar with this subfield, but in so far as I can tell, this work seems very novel as it seems to be the first work to introduce $\mu$P for local learning. The authors provide theoretical guarantees showing that it is possible to reduce the required amount of hyperparameters in local learning.

Quality: The theoretical results are well-justified by detailed proofs in the appendix and references to prior works. The authors provide empirical justification on real-world datasets as well.

Significance: The paper is significant as it provides justification on why local learning can match performance of models trained by back-propagation.

**Weaknesses:**

The content is rather inaccessible as the assumed knowledge is too specific for a broad conference such as ICLR. I have background in mean-field networks, but not local learning or $\mu$P. It would be useful to add a brief appendix on these as a background (beyond what is already described in Section 3) to help the reader understand the paper.

The paper is also hard to follow due to the amount of abbreviations. It would also be useful to add backreferences for the techniques F-ini and FPA in lines 158, 162.

Some of the theorems are missing assumptions, notation, or intuition. For example, in Prop. 3.3 (line 250), what are $\delta_l$, $e_L$? In Cor. 4.3 (line 350), why do we need $a_L + b_L = 1$?

The experimental results are on relatively small datasets, e.g. FashionMNIST and CIFAR10: it would be nice to have validation on CIFAR100 as well.

**Questions:**

Line 200: what is abc-parametrization?

Line 278: what is $M^{\overline{\gamma}_L}$?

Line 309: what is $\mu$Transfer? A reference here would be helpful.

What would the challenges be in extending the analysis to non-linear networks?

[HRSS19 Mean-field Langevin dynamics and energy landscape of neural networks, arXiv preprint arXiv:1905.07769] studies infinite-width neural networks via the lens of mean-field Langevin dynamics. Would it be possible to study local learning through this lens, and could this potentially mitigate the requirement to develop a tensor program (line 535)?

---

> ### Author Response · Authors · 2024-11-23
> **Response to Reviewer ubuK**
>
> We would like to thank the reviewer for the helpful comments and suggestions.
>
> ### Weaknesses
>
> > It would be useful to add a brief appendix on these as a background (beyond what is already described in Section 3) to help the reader understand the paper.
>
> Thank you for the valuable advice. We have added an extended background section to Appendix A.1, providing detailed explanations of $\mu$P, Predictive Coding (PC), and Target Propagation (TP). Specifically, we focused on the following two points:
>
> 1.	We introduced a more detailed notation for PC and TP, along with their historical backgrounds. This addition clarifies the motivation for considering these methods.
> 2.	We provided an overview of subsequent research on $\mu$P, making it easier to understand why $\mu$P possesses desirable properties such as $\mu$Transfer.
>
> We believe these additions have improved the clarity and accessibility of our paper.
>
> > The paper is also hard to follow due to the amount of abbreviations. It would also be useful to add backreferences for the techniques F-ini and FPA in lines 158, 162.
>
> Thank you for the helpful suggestion.
> Following your suggestion, we have added a table summarizing the abbreviations of technical terms to assist general readers (Table S.1).
> we have also added backreferences for F-ini and FPA, that is, we have added hyperlinks to these terms, linking them to the paragraph where these techniques are first introduced.
> This will improve the readability of the paper.
>
> > in Prop. 3.3 (line 250), what are $\delta_l$ and $e_L$? In Cor. 4.3 (line 350)
>
> Thank you for your question. In Proposition 3.3, $\delta_l$ represent $\delta_l = \nabla_{u_l} u_L$, as defined in Section 3.1.2. Additionally, $e_L$ is consistently defined as $e_L = y - h_L$ regardless of learning methods.
>  We have added definitions for these symbols directly in Proposition 3.3. Furthermore, we have modified instances of ${e}_L$ to $\text{diag}({e}_L)$, making the notation more applicable for cases with more than one class.
>
> > In Cor. 4.3, why do we need $a_L+b_L=1$?
>
>  The theme of Section 4.2 (including Cor. 4.3) is feature learning and we suppose the $\mu$P setting.  To clarify this, we have added the statement: “the weight is initialized with $\mu$P, i.e., $a_L + b_L = 1$.”
> As a side note, when initialized with $a_L + b_L = 1/2$ (the kernel regime), the preconditioner component $C_\gamma$ remains of order 1, even if $\gamma_{L} = \Theta(1)$.
>
> Intuitively, the shift between first-order and second-order-like gradients in Cor. 4.3 comes from a change in how effectively the weights in the last layer propagate to the lower layers.
> Similar to linear regression, the inference solution of the last layer inherently involves an inverse matrix and its solution is similar to GNT.
> Decreasing $a_L + b_L$ moves the gradient closer to GNT since increasing the initial weight scale of the last layer amplifies the impact of the last inference solution on the lower layers.
> On the other hand, increasing $a_L + b_L$ reduces the scale of the weights in the final layer, thereby decreasing the influence of the last layer. Furthermore, by increasing $a_L + b_L$ and simultaneously decreasing $\gamma_L$, which represents how effectively the error from the last layer propagates downward, the gradients become more BP-like.
> Conversely, when $a_L + b_L$ is small or $\gamma_L$ is large, the influence of the inference solution from the final layer becomes stronger, making gradients more similar to GNT.
>
> ### Questions
>
> > Line 200: what is abc-parametrization?
>
>  The abc-parameterization $\{a_l, b_l, c_l\}_{1 \leq l \leq L}$ determines the scaling of weights and learning rates at initialization. We have added an explanation of the meaning of this definition for clarity.
>
> > Line 278: what is $M^{\bar{\gamma}_L}$?
>
> $\bar{\gamma_L}$ is introduced as an extension of the abc-parameterization and indicates how $\gamma_L$ should be scaled, similar to the scaling of $\{a_l, b_l, c_l\}_{1 \leq l \leq L}$. Specifically, when $\bar{\gamma}_L = 0$, ${\gamma}_L$ is kept constant across the width, whereas when $\bar{\gamma}_L = -1$, ${\gamma}_L$ is scaled proportionally to the width.
>
> > Line 309: what is $\mu$Transfer? A reference here would be helpful.
>
> Thank you for your suggestion.
> In Yang et al. 2021, $\mu$Transfer is defined by satisfying both “we can set the optimal learning rate without depending on the order of width” and “wider is better.”
> More specifically, $\mu$Transfer in learning rates refers to the ability to reuse $\eta'$ in $ \eta_l = \eta_l'/M^{c_l}$ (Eq. 9 in the main text) across different widths.
> We have clarified this definition in the indicated section and added a reference for further context.
>
> [1] Yang et al. Tuning large neural networks via zero-shot
> hyperparameter transfer. In Advances in Neural Information Processing Systems, 2021.

---

> > ### Author Response · Authors · 2024-11-23
> > **Response to Reviewer ubuK (Part2)**
> >
> > > Challenges in extending the analysis to non-linear networks
> >
> > In the derivation of $\mu$P, which typically assumes an infinitesimal one-step update, the nonlinearity can be handled through complex but straightforward calculations (as is shown in previous work on SGD and our calculation on PC). This is because, essentially, the order evaluation reduces to that of neural networks at random initialization by the first-order Talyor expansion, and such analyses on random neural networks are well established (e,g., Yang 2020).
> >
> > Essentially, we think that the challenges for nonlinear cases are analyses of dynamics (e.g., the solvability of dynamic equations, convergence rate, and fixed-point analysis). In our study, Theorem 4.2, which is the fixed point analysis, essentially seems non-trivial to extend to the nonlinear case.
> > However, it will be noteworthy that even in linear networks, analyzing dynamics is highly non-trivial and challenging. This is because the dynamics are nonlinear (nonlinear differential equations) even in linear networks, and convergence properties remain unclear in general. These aspects are still being studied under various conditions, for recent examples,
> >
> > [2] Kunin et al., Get rich quick: exact solutions reveal how unbalanced initializations promote rapid feature learning, NeurIPS 2024
> > [3] Tu et al., Mixed Dynamics In Linear Networks: Unifying the Lazy and Active Regimes, NeurIPS 2024
> >
> > Our theorem is not on the learning dynamics but on the inference dynamics, but the essential point is the same between them.
> > Thus, we believe that obtaining a non-trivial fixed point can be regarded as solving a somewhat challenging problem in linear networks.
> >
> > > Mean-field Langevin dynamics ... would it be possible to study local learning through this lens, and could this potentially mitigate the requirement to develop a tensor program
> >
> > Although this is beyond the scope of our paper, we agree that exploring mean-field dynamics will be an exciting next step in understanding the learning dynamics of nonlinear (shallow) networks.
> >  In particular, the mean-field Langevin dynamics you mentioned can realize efficient convergence complexity.
> >
> > The problem, however, seems to lie in local loss optimization; constructing the weight distribution $ \rho_t(w_{2}, w_{1})$, with $w_{l}$ denoting the weight vector of the $l$-th layer and $t$ the time step, becomes highly non-trivial. For instance, in PC, the inference update requires tracking the dynamics of the state $h_1$.  It will demand us to evaluate the conditional distribution $\rho_t(w_{2}, w_{1}|h)$ during the learning phase and  $\rho_t(h|w_{2}, w_{1})$ during the inference phase, which seems an entirely novel challenge for the mean-field framework.
> > In TP, forward and feedback networks are learned simultaneously
> > and we will need to evaluate interaction like $\rho_t(w_{2}, w_{1}|Q_1, Q_2)$ and  $\rho_t(Q_1, Q_2|w_{2}, w_{1})$ where $Q_l$ denotes feedback networks' weights. We hope future research will take on these exciting challenges.
> >
> > We believe mean-field dynamics hold great potential for providing qualitative insights into learning, particularly in shallow networks, compared to the tensor program.
> > However, it would be important to note that tensor programs are more versatile for proving the existence of infinite-width dynamics in broader settings, including networks of any depth, various activation functions, and step and batch sizes. In this sense, depending on the objective, both approaches are valuable and complement one another.

---

> > > ### Comment · Reviewer_ubuK · 2024-11-25
> > > **Thank you for your response.**
> > >
> > > Thank you for addressing my questions. In light of the update, I will increase my score.

---

### Official Review · Reviewer_Nzx8 · 2024-11-04

**Soundness:** 3
**Presentation:** 2
**Contribution:** 3
**Rating:** 8
**Confidence:** 2

**Summary:**

This work studies parameterisation for local learning algorithms: for predictive coding and target propagation. The main goal is to understand maximal update parameterization ($\mu P$), which allows transfer of hyperparameters across different widths. Authors prove optimal $\mu P$ from one step of descent or for linear networks. Experimental results are provided based on Fashion MNIST.

**Strengths:**

Authors give a thorough analysis of parametrization in local learning, motivated by previous progress of $\mu P$ in backpropagation. The question is well-motivated by the fact that model sizes for local learning are increasing, thus it is important to understand which parameterizations are robust to scaling up the models.

Experimental results on FashionMNIST confirm theoretical findings.

**Weaknesses:**

Presentation of the paper can be improved, in particular the notation section. Choice of $\mathcal{L}$ is not introduced, notation for $\gamma_l$ should be defined before Section 3.1. Also, $\bar \gamma_L$ is an important quantity, but is only defined implicitly in Theorem 4.1.

Both sections on PC and TP were a bit hard to follow (for someone outside of the field): In PC, for example, Equation (3) should also be complemented by the case $\ell = L - 1$.

In Table 1, adding references to prior work and highlighting new contributions will be helpful.

**Questions:**

How is $\eta'_l$ chosen in Equation (8)? Is it tuned in the experiments?

Figure 3: what is the cause of sharp loss increase in the loss of wide networks when $\bar \gamma_L < -1$? Is there some theoretical explanation? Can be it that for $\bar \gamma_L = -1$, training would also diverge for wide enough network?

---

> ### Author Response · Authors · 2024-11-23
> **Response to Reviewer Nzx8**
>
> We would like to thank the reviewer for the helpful comments and suggestions.
>
> ### Strengths
>
> > Experimental results on FashionMNIST confirm theoretical findings.
>
> Thank you for your kind feedback.
> While we mainly showed figures on FashionMNIST in the main text, we also demonstrated the results on CIFAR-10 in the appendix. Additionally, based on the comments by Reviewer ubuK, we have added experiments using CIFAR-100 in this rebuttal.
>
> ### Weaknesses
>
> > Choice of $\mathcal{L}$ is not introduced
>
> Thank you for catching this.
>  $\mathcal{L}$ denotes the loss function (for the output layer).
>  The derivation of $\mu$P by one-step updates allows continuously differentiable ones in general, but usually mean squared loss (MSE) or cross-entropy (CE) losses are intensively argued. Similarly, PC and TP studies usually use MSE or CE, and thus we assume MSE or CE in this work.
> More, specifically, when the CE loss is used in Eq. S29 instead of MSE loss, $\delta_L = y - t$ becomes $\delta_L =y - \text{softmax}(t)$, and the order analysis remains unchanged.
> Note that we have also empirically verified the learning rate transfer with CE loss, as shown in Figure S.5 (Appendix D.1.2).
> Since there was no reference to this point in the main text, we have now added a mention to address it explicitly in Line 322 and in the caption of Figure 4.
>
> > notation for $\gamma_l$ should be defined before Section 3.1. Also, $\bar{\gamma_L}$ is an important quantity, but is only defined implicitly in Theorem 4.1.
>
> Thank you for your feedback.
> We have now introduced $\bar{\gamma_L}$ in preliminaries Section 3.1.1 for better clarity.
> Regarding the notation for $\gamma_l$, we plan to retain its explanation in Section 3.1, as this is where the mathematical formulation begins. However, to clarify its role, since it was not well-explained in Section 3.1.1, we have added a note after Equation (3) stating, “$\gamma_l$ can also be regarded as a step size during the inference phase.”
> Thank you for the excellent suggestion.
>
> > Both sections on PC and TP were a bit hard to follow (for someone outside of the field): In PC, for example, Equation (3) should also be complemented by the case $l=L-1$
>
> Thank you for your advice. To improve clarity, we created an extended background section in Appendix A, where we added detailed explanations and notations for PC and TP.
> Additionally, as you suggested, it is clearer to treat the case of $l = L - 1$ separately in Equation (3). We have addressed this separately in Section A.1.2.
>
> > In Table 1, adding references to prior work and highlighting new contributions will be helpful.
>
> Thank you for your advice.
> We have added citations to clarify the sources and highlighted our new contributions in bold with the label “New”.
>
> ### Questions
>
> > How is $\eta_l'$ chosen in Equation (8)? Is it tuned in the experiments?
>
> Thank you for your question.
> From the theoretical perspective of infinite width, $\gamma'$ is an uninteresting constant that does not depend on the width.
> However, from a practical perspective, $\eta_l'$ is tuned by experiments. For example, in experiments such as those shown in Figures 1 and 4 (graphs for $\mu$Transfer of learning rate), the x-axis labeled as “LR” represents $\eta_l'$. The graphs illustrate the behavior of $\eta_l;$ across models with varying widths.
> Furthermore, by using $\mu$Transfer, it is possible to reduce the cost of learning rate tuning for large models by reusing the $\eta_l'$ tuned on smaller models to larger models.
> Note that this tuning of $\eta_l'$ is common among $\mu$P studies.
>
> > Figure 3: what is the cause of sharp loss increase in the loss of wide networks when $\bar{\gamma}_L<-1$?  Is there some theoretical explanation? Can be it that for $\bar{\gamma}_L=-1$
>
> Thank you for your question.
> Yes, we can explain the sharp increase in loss for $\bar{\gamma_L} < -1$ theoretically.
> This is because the inference is unstable for $\bar{\gamma_L} < -1$. Specifically, for $\bar{\gamma_L} < -1$, $\Delta u_{l}$ grows with the network width, making the inference phase unstable.
> The largest value of $\gamma_L$ where the loss does not diverge with increasing width is $\bar{\gamma_L} = -1$, where inference is stable. In this case, $\Delta u_{l}$ remains constant in terms of width, ensuring that learning does not become unstable even as the network width increases.

---

> > ### Comment · Reviewer_Nzx8 · 2024-11-25
> >
> > Thank you for addressing my comments. I noticed that the caption of Figure 2 is misaligned with figures (a, b, c). Related to your last point: Could you also provide / point me to the proof for Corollary 4.3? I cannot find it in the current version, and it does not seem obvious to me.

---

> > > ### Author Response · Authors · 2024-11-27
> > >
> > > > I noticed that the caption of Figure 2 is misaligned with figures (a, b, c).
> > >
> > > Thank you for catching this.
> > > We have revised the caption of Figure 2.
> > >
> > > > Could you also provide/point me to the proof for Corollary 4.3? I cannot find it in the current version, and it does not seem obvious to me.
> > >
> > > Thank you once again for your helpful feedback!
> > > We agree that it would be more reader-friendly to explicitly include the proof of this corollary. Thus, we have added the proof in Appendix B.2.2. Please recall that this corollary assumes random initialization given by $\mu$P. Due to this assumption, the derivation follows straightforwardly from Theorem 4.2.

---

> > > > ### Comment · Reviewer_Nzx8 · 2024-11-27
> > > >
> > > > I would like to thank the authors for addressing all my concerns and improving presentation of the paper. I am raising my score.

---

### Official Review · Reviewer_SFwu · 2024-11-04

**Soundness:** 3
**Presentation:** 2
**Contribution:** 3
**Rating:** 6
**Confidence:** 2

**Summary:**

The submission presents derivation of the initialization and step-size scaling required to achieve hyperparameter transfer across width using the $\mu P$ parameterization in the infinite width limit for two local learning algorithms, predictive coding and target propagation.

**Strengths:**

To my knowledge, the results are new as the infinite width limit has not received as much attention for local learning as for the typical backpropagation-based methods. The objectives of the paper are clear and the paper executes on its premises. The experimental results support the claim that the derived parameterization does achieve hyperparameter transfer.

**Weaknesses:**

The paper deals with two very specialized subjects, local learning and infinite-width limits. As a result, it has a difficult job in making its argument clear to a general audience. The current writing makes a reasonable attempt, but there are still points that can be clarified to help guide the reader that might be only be vaguely familiar with one of the topic. I give a list of specifics in the questions below.

**Questions:**

The discussion of the implications of the $\mu P$ parameterization for PC after corollary 4.3 and in §4.3, on the impact of the free parameter $\gamma$, is very technical. It is not clear what the broader implications of $\gamma = 0$ or $\gamma = 1$ or the fact that the the PC gradients are more similar to those of standard backprop GD or GNT. A higher-level description of what the parameter $\gamma$ represents, and what it means to be closer to 0 or 1, would help the reader grasp the importance of the results.

- It would be worth to explicitly state that, by definition, $\mu P$ depends on the algorithm used, and therefore needs to be derived to figure out what $\mu P$ means in the context of PC and TP, which is the contribution of this paper. The sentence before the list of contributions leaves the "$\mu P$ depends on the algorithm" part implicit.

- The word "inference" is heavily overloaded in ML and a clarifying note as to what it means for PC in §2 would help. This is adressed later in §3.1 but a short parenthetical might be worth including.

- "The parameterization that induces the feature learning regime is called $\mu P$"
  This phrasing implies that there is a unique parameterization that induces feature learning. It seems that the authors instead mean that "$\mu P$ induces the feature learning regime"?

- "The training dynamics are referred to as stable when they neither vanish nor explore as the network width increases"
  What quantitiy should neither vanish nor explode?

- "the parameterization that achieves this [non-vanishing and non-exploding] is called a stable parameterization"
  The phrasing as "the parameterization that achieves this" implies that this parameterization is unique. Is this what is meant or should it be "a"?

- In table 1; reminding the reader of what the parameters $b_l$, $c_l$ and $\gamma_l$ would help, e.g. $b_l: initialization of the weights, $c_l$: scaling of the step-size. Adding a legend would help remind the reader of what SP, SGD, GNT, PC and TP mean.

- Line 271-272 "the $\mu P$ of BP can be directly transferred to PC" should also mention "and leads to parameter transfer across width", as this is what the figure shows.

- Please add that $\theta_l = 2a_l + c_l$ to the statement of Thm 4.1, assuming that those parameters indeed have the same meaning as in Prop 3.3.

- An additional experiment showing that the $\mu P$ for SGD does not lead to hyperparameter transformer on PC without FPA would strenghten the claim that the $\mu P$ for PC is necessary

- It is not clear whether the difference between GD, (S)GD and SGD is intended to meaningful or whether they are used interchangeably. If it is not, would it make sense to standardize to GD or (S)GD?

- Figure 2 should explicitly state what is being compared; The cosine similarity measures the similarity between what and what?

---

> ### Author Response · Authors · 2024-11-23
> **Response to Reviewer SFwu (Part1)**
>
> We would like to thank the reviewer for the helpful comments and suggestions.
>
> > The paper deals with two very specialized subjects, local learning and infinite-width limits. As a result, it has a difficult job in making its argument clear to a general audience.
>
> We agree that this work bridges two different domains (infinite-width and local learning), and more detailed explanations could be helpful for general readers to cover the background.
> To address this, we have added an extended background section to Appendix A.1, following the advice by reviewer ubuK, providing detailed explanations of $\mu$P, Predictive Coding (PC), and Target Propagation (TP).
>
> > A higher-level description of what the parameter represents, and what it means to be closer to 0 or 1, would help the reader grasp the importance of the results.
>
> Thank you for your valuable feedback. We guess that what you mention is the problem of $\bar{\gamma}_L=0$ or $-1$.
> When $\bar{\gamma}_L$ is closer to -1, $\gamma_L$ is larger than other $\gamma$ values, and errors from the last layer are propagated more effectively.
> In linear regression with mean squared loss, the solutions involve inverse matrices. Similarly, in our setting, the inference solution for the representation vector of the last layer also involves inverse matrices. Thus, when $\bar{\gamma}_L$ is closer to $-1$, the representation vector of the last layer is computed first and then propagated to the lower layers, resulting in a behavior reminiscent of the Gauss Netwon Target.
> Some prior studies empirically reported this appearance of  Gauss Netwon Target, and our theory is the first work that gives solid theoretical backing.
> Furthermore, we found that differences in the order of $\gamma$ universally switch the behaviors of Gradient Descent (GD) and Gauss Netwon Target (GNT).
> We have included this intuitive explanation in the main text.
>
> > The sentence before the list of contributions leaves the "$\mu$P  depends on the algorithm" part implicit.
>
> Thank you very much for the valuable advice. Following your suggestion, we have added the statement “$\mu$P depends on the algorithm” before the list of contributions.
>
> > The word "inference" is heavily overloaded in ML and a clarifying note as to what it means for PC in §2 would help. This is addressed later in §3.1 but a short parenthetical might be worth including.
>
> Thank you very much for the valuable advice. Certainly, “inference” is used with multiple meanings within the ML community, which may have made the description in §2 unclear. In Predictive Coding, “inference” refers to the process of inferring the appropriate activations for hidden layers during the inference phase. To clarify this, we have added a brief explanation in §2 (when “inference phase” first appears).
>
> > "The parameterization that induces the feature learning regime is called $\mu$P " This phrasing implies that there is a unique parameterization that induces feature learning. It seems that the authors instead mean that "$\mu$P induces the feature learning regime"?
>
> Thank you for your feedback. Since the original text may be unclear to a general audience, we have rewritten it in the manner you suggested. Actually, Yang et al 2021 claims that  $\mu$P is a unique parameterization that induces feature learning under the stability of learning, conditions 1 and 2, for SGD.
> These conditions seem rational for characterizing feature learning in the infinite width, and, in that sense, $\mu$P is unique.
> To avoid misunderstanding, we revised the manuscript to clarify in what sense $\mu$P is unique.
>
> > "The training dynamics are referred to as stable when they neither vanish nor explore as the network width increases" What quantity should neither vanish nor explode?
>
> Thank you for your feedback. We can say an abc-parameterization is stable if, for $l<L$ and for any fixed $t \geq 1$,
> $\Delta h_{l,t} = O(1), \ \Delta f_{t} = O(1)$ (please see the precise definition given in Definition A.5).
>
> > "the parameterization that achieves this [non-vanishing and non-exploding] is called a stable parameterization" The phrasing as "the parameterization that achieves this" implies that this parameterization is unique. Is this what is meant or should it be "a"?
>
> We guess that our answers to your above two feedbacks have solved this concern.
> The original work of $\mu$P refers to it as a unique stable parameterization under specific conditions, and  [non-vanishing and non-exploding] is a simpler paraphrase of the stability definition (Definition A.2).
>
> [1] Yang et al. Tuning large neural networks via zero-shot hyperparameter transfer. In Advances in Neural Information Processing Systems, 2021.

---

> ### Author Response · Authors · 2024-11-23
> **Response to Reviewer SFwu (Part2)**
>
> > In table 1; reminding the reader of what the parameters $b_l, c_l$ and $\gamma_l$ would help, e.g.$b_l$:initialization of the weights, $c_l$: scaling of the step-size. Adding a legend would help remind the reader of what SP, SGD, GNT, PC and TP mean.
>
> Following your suggestion, we have updated the caption of Table 1 to clarify what $b_l$, $c_l$, and $\gamma_l$ represent. This has indeed improved the clarity of Table 1.
>
> > Line 271-272 "the $\mu$P of BP can be directly transferred to PC" should also mention "and leads to parameter transfer across width", as this is what the figure shows.
>
> Thank you for the advice. We have added the phrase “and leads to parameter transfer across width” at the end.
>
> > Please add that $\theta_l=2a_l+c_l$ to the statement of Thm 4.1, assuming that those parameters indeed have the same meaning as in Prop 3.3.
>
> We have added the statement $\theta_l = 2a_l + c_l$ in Theorem 4.1 as well. This has made the statement of Theorem 4.1 clearer.
>
> > An additional experiment showing that the $\mu$P for SGD does not lead to hyperparameter transformer on PC without FPA would strengthen the claim that the $\mu$P for PC is necessary
>
> Thank you for the suggestion.
> Firstly, in Figure1, $\mu$P for PC at $\bar{\gamma}_L = 0$ corresponds to $\mu$P for SGD, and thus $\mu$P for SGD also achieves learning rate transfer in PC in this figure.
> However, in the following two cases, $\mu$P for SGD fails to achieve learning rate transfer:
>
> 1. Stability during inference is particularly important when the heuristics F-ini is not assumed in addition to FPA. Figure 4 shows the learning rate transfer without FPA and F-ini is shown in Figure 4. We find that $\mu$P($\bar{\gamma}_L = 0$) i.e. $\mu$P for SGD is trivial (the inference loss does not decrease as the width increases) and the $\mu$Transfer for learning rate does not hold.
> 2. Even for PC without FPA and with F-ini, if we use base width, $\mu$P($\bar{\gamma}_L = -1$) may be practically advantageous over $\mu$P for SGD as shown in Figure S.10.
>
> In these cases, at $\mu$P for SGD, the $\mu$Transfer for learning rate does not hold and the $\mu$P for PC (specifically, $\mu$P at ($\bar{\gamma}_L = -1$)) is necessary.
>
> > It is not clear whether the difference between GD, (S)GD and SGD is intended to meaningful or whether they are used interchangeably. If it is not, would it make sense to standardize to GD or (S)GD?
>
> Thank you very much for the insightful feedback.
> Since $\mu$P can be derived in one step, it is capable of handling both cases, as stated in the original $\mu$P paper [Yang and Hu 2021]. Following the original paper, we have decided to consistently use the term SGD throughout.
> In Section 4.2, we focus on inference over a single weight update step, and in this section, we use the terms "first-order GD" or simply "GD" to emphasize that multiple mini-batches are not used.
>
> > Figure 2 should explicitly state what is being compared; The cosine similarity measures the similarity between what and what?
>
> Thank you for the valuable advice. In Figure 2, we measured the cosine similarity between the gradients analytically derived in Theorem 4.2 and the BP gradients or GN gradients for each layer. We have added this description to the caption.

---

> > ### Comment · Reviewer_SFwu · 2024-12-02
> >
> > I thank the authors for their response and the added clarifications, these will make the submission more accessible.

---

### Official Review · Reviewer_akLS · 2024-11-06

**Soundness:** 3
**Presentation:** 3
**Contribution:** 3
**Rating:** 8
**Confidence:** 1

**Summary:**

This paper investigates local learning methods for neural networks as alternatives to backpropagation. It specifically focuses on predictive coding (PC) and target propagation (TP). The authors introduce maximal update parameterization (µP) for PC and TP in the infinite-width limit to enable hyperparameter transfer across different model widths, aiming to stabilize local loss optimization. They analyze deep linear networks to show that PC’s gradient transitions between first-order and Gauss-Newton-like gradients depending on parameterization. For TP, they find that µP favors feature learning over the kernel regime, a distinction from traditional BP.

**Strengths:**

- The paper is well-written.
- The problem is conceptually well motivated.
-  The manuscript includes both rigorous proofs and empirical experiments that validate the findings.
- The observation that the optimal learning rate does not depend on the order of width is interesting.
- The disappearance of the kernel regime in TP is interesting.

**Weaknesses:**

1. The derivation of $\mu P$ assumes one-step gradient
2. The derivation also assumes linear networks
3. It is unclear how the findings extend to more general settings

**Questions:**

1. In the preliminaries I can't find any definition for $\mathcal{L}$, I assume it is the loss function, are there any particular assumptions about it?
2. A precise definition of the Maximal Update Parametrization in the text would be useful, especially for readers unfamiliar with the related work.
3. In equation 5, where is $\psi$ defined?
4. Could the authors provide a more precise definition of stable training dynamics?
5. Is the transfer of learning rates also observed for Cross Entropy loss?
6. How sensitive are $\mu P$  and $\mu$Transfer to different initialization schemes?

---

> ### Author Response · Authors · 2024-11-23
> **Response to Reviewer akLS**
>
> We appreciate the reviewer's helpful comments and suggestions.
>
> > In the preliminaries I can't find any definition for $\mathcal{L}$, I assume it is the loss function, are there any particular assumptions about it?
>
> Thank you for catching this. Yes, $\mathcal{L}$ denotes the loss function (for the output layer).
>  The derivation of $\mu$P by one-step updates allows continuously differentiable ones in general, but usually mean squared loss (MSE) or cross-entropy (CE) losses are intensively argued. Similarly, PC and TP studies usually use MSE or CE, and thus we assume MSE or CE in this work.
> Note that Theorem 4.2 (fixed point analysis of PC) is an exception. In this theorem, we explicitly derive a closed-form solution for inference and this is not obvious for the CE. We have clarified these assumptions in the revised manuscript.
>
> > A precise definition of the Maximal Update Parametrization in the text would be useful, especially for readers unfamiliar with the related work.
>
> Thank you for your feedback. The original work of $\mu$P determines an abc-parameterization satisfying the stability of learning, conditions 1 and 2 (Theorem 5.6 of [Yang and Hu 2021]) as $\mu$P. We added this precise definition as Definition A.5.
> We believe that this will further clarify the meaning of $\mu$P for general readers.
>
> > In equation 5, where is $\psi$ defined?
>
> Although we explain the details of the activation function $\psi$ in Appendix A.1, we agree that it will be better to remark in the main text. In practice, the same activation function as one used in the forward pass is often utilized (i.e., $\psi = \phi$).
> We set $\psi = \phi$ in experiments and assume $\psi$ as the identity function only in Theorem 5.1.
>
> > Could the authors provide a more precise definition of stable training dynamics?
>
> Thank you for catching this. Although we remarked the original definition of stability in the middle of the derivation of $\mu$P in the appendix (that is,  Definition A.2), we did not mention its detail in the main text.
> Following your suggestion,  we have revised the manuscript to explicitly mention this definition and also enriched the explanation of its meaning in the Appendix A.2.
>
> > Is the transfer of learning rates also observed for Cross Entropy loss?
>
>  Yes, we have also empirically verified the learning rate transfer with Cross Entropy loss in Figure S.5 (Appendix D.1.2). Since there was no reference to this point in the main text, we have now added a mention to address it explicitly.
>  It is also noteworthy that the $\mu$P (including ours) can be directly applied to the CE loss.
> Specifically, when the CE loss is used in Eq. S29 instead of MSE loss, $\delta_L = y - f$ becomes $\delta_L =y - \text{softmax}(f)$, and the order analysis remains unchanged.
>
> > How sensitive are $\mu$P and $\mu$Transfer to different initialization schemes?
>
> In addition to $\mu$P, we trained with SP (PyTorch Default) and NTK parameterizations. From our experiments, the optimal learning rate depends on the width when trained with SP. Therefore, the parameterization is sensitive for $\mu$Transfer.
> Furthermore, we conducted experiments on the NTK parameterization in the Appendix. While in the NTK parameterization, the optimal learning rate does not depend on width, its performance deteriorates as the width increases, which is due to approaching the kernel regime. Consequently, $\mu$P is unique in achieving high performance at large widths while maintaining a width-independent learning rate.
>
> > A comment on weaknesses: the derivation of $\mu$P assumes one-step gradient. The derivation also assumes linear networks.
>
> Thank you for your comment. As we mentioned in the Limitation paragraph in the last section, the assumption of one-step gradient (for $\mu$P derivation in non-linear networks) and that of linear networks (for the dynamics of general time steps) are not unique to our work
> but common among $\mu$P work [e.g., Yang \& Hu 2021, Yang et al., 2024].
> Although the generalization of theories itself is an intriguing challenge in the field,
> this study has provided interesting and novel insights into PC and TP, confirming it is a useful approach even under the current assumptions.
>
> [1] Yang et al. Tuning large neural networks via zero-shot hyperparameter transfer. In Advances in Neural Information Processing Systems, 2021.
> [2] Yang et al.. Feature learning in infinite-depth neural networks. In International Conference on Learning Representations, 2024

---

> > ### Comment · Reviewer_akLS · 2024-11-28
> >
> > Thank you for thoroughly addressing my questions and providing clear clarifications. Although I’m not deeply familiar with this specific area, your work appears solid and well-executed, which has led me to raise my score. I appreciate the effort you’ve put into this research and wish you best of luck!

---

### Author Response · Authors · 2024-11-23
**We thank all the reviewers again for their valuable feedbacks.**

We are pleased to see that all reviewers, after their careful reading, have provided an overall positive assessment of our contributions and acknowledged the significance. Some reviewers suggested that improving the writing on background knowledge and technical terms (or jargon) would help a broader audience understand. In response, we have added more detailed and **accessible explanations of definitions, technical terms, and extended background information**, tailored for a general audience. In particular, we hope that **Section A in the Appendices** will prove helpful. We would greatly appreciate it if you could look over our revised manuscript and consider reevaluating your scores.

Our research bridges infinite-width analysis and local learning, offering novel and intriguing insights for both domains. We believe that our work lays a solid foundation to engage these communities and promote further developments in subsequent research.

---

### Meta-Review · Area_Chair_cfyr · 2024-12-16

**Metareview:**

The paper presents theoretical advances towards better under of local learning. The specific focus is on parameterization of local learning, for predictive coding (PC) and target propagation (TP). Based on maximal update parameterization, the paper shows that hyper parameters can be transferred across different widths, similar to related results for backpropagation.  Focusing on deep linear networks, the paper shows specific results for PC and TP. The reviewers generally appreciated the advances made in the paper, as the results are mostly clearly presented. There were some concerns regarding the specific assumptions, such as linear networks, and more broadly on the quality of the exposition, as arguably a specialized audience has knowledge of these advances. The authors have added a section in the append

**Additional Comments On Reviewer Discussion:**

The reviewers engaged with the authors during the discussion phase, acknowledging their responses and at time asking follow-up questions to which the authors responded.

---

### Decision · Program_Chairs · 2025-01-22

Accept (Poster)